# Epac1 increases myosin regulatory light-chain phosphorylation, energetic cost of contraction, and susceptibility to heart failure

Yoshiki Ohnuki[1,2], Kenji Suita[1,2], Misao Ishikawa[3], Yasumasa Mototani[1],
Megumi Nariyama[4], Aiko Ito[5], Ichiro Matsuo[1,2,6], Yoshio Hayakawa[7], Akinaka Morii[1,6],
Takao Mitsubayashi[1], Yasutake Saeki[1], Yoshihiro Ishikawa[2], Satoshi Okumura[1,2,*]

1 Department of Physiology, Tsurumi University School of Dental Medicine, Yokohama, Japan,
2 Cardiovascular Research Institute, Yokohama City University Graduate School of Medicine,
Yokohama, Japan, 3 Department of Oral Anatomy, Tsurumi University School of Dental Medicine,
Yokohama, Japan, 4 Department of Pediatric Dentistry, Tsurumi University School of Dental Medicine,
Yokohama, Japan, 5 Department of Orthodontics, Tsurumi University School of Dental Medicine,
Yokohama, Japan, 6 Department of Periodontology, Tsurumi University School of Dental Medicine,
Yokohama, Japan, 7 Department of Dental Anesthesiology, Tsurumi University School of Dental
Medicine, Yokohama, Japan

* okumura-s@tsurumi-u.ac.jp

journal.pone.0325986

KOREA, REPUBLIC OF

**Peer Review History:** PLOS recognizes the
benefits of transparency in the peer review
process; therefore, we enable the publication
of all of the content of peer review and
author responses alongside final, published
articles. The editorial history of this article is
available here: https://doi.org/10.1371/journal.
pone.0325986

## Abstract

β-Adrenergic receptor (β-AR) stimulation of the heart, leading to increased cardiac
output, is mediated by cyclic AMP (cAMP), which induces protein kinase A (PKA)-
mediated phosphorylation of the myofilament proteins troponin I (TnI) and myosin
binding protein-C (MyBP-C). The aim of this study was to investigate the contribu-
tion of the exchange protein activated by cAMP (Epac1), a PKA-independent cAMP
effector, to the response of cardiac myofilaments to β-AR stimulation. The calcium
sensitivity of force and ATPase activity, and the tension cost (ATPase activity/force)
were significantly greater in skinned myocardium from transgenic mice specifically
overexpressing Epac1 in the heart (Epac1TG) and wild-type (WT) mice treated with
8CPT-AM, an Epac-selective cAMP analogue, as compared with non-transgenic
(NTG) or control mice, respectively. In addition, myosin regulatory light chain (RLC)
phosphorylation was significantly greater in Epac1TG and WT mice treated with
8CPT-AM than in NTG or control mice via phospholipase C/phosphokinase C,
without any change in the phosphorylation of TnI or MyBP-C. We also examined the
effects of chronic β-AR stimulation on cardiac function in Epac1TG. The left ven-
tricular ejection fraction was significantly decreased from baseline in both NTG and
Epac1TG after isoproterenol infusion (60 mg/kg/day for 1 week), but the magnitude of
the decrease was much greater in Epac1TG. Our results suggest that Epac1 acti-
vation might induce an imbalance between force-generating capacity and ATPase
activity in skinned myocardium. This could increase oxygen consumption and the

**Data availability statement:** All data are in the paper and Supporting Information files.

**Funding:** This study was supported by the Japan Society for the Promotion of Science (JSPS) KAKENHI Grant (17K12067, 20K10304, 23K09517 to Dr. Yoshiki Ohnuki, 20K10305, 23K09493 to Dr. Kenji Suita, 22K10255 to Dr. Megumi Nariyama, 19K24109, 21K17171 to Dr. Aiko Ito and 21K10242, 24K13250 to Dr. Satoshi Okumura). The founders had no role in study design, data collection and analysis, decision to publish, or preparation of the manuscript.

**Competing interests:** The authors have declared that no competing interests exist.

energetic cost of contraction in living myocardium under conditions of chronic β-AR stimulation, leading to the development of heart failure.

## Introduction

Intracellular calcium ($Ca^{2+}$) was identified as a trigger of muscle contraction in 1962 [1]. Although force generation can be achieved with only myosin and actin, additional proteins are required for its regulation. Subsequent research to understand the regulatory mechanisms focused mainly on the roles of thin-filament-associated proteins, such as troponin I (TnI), troponin T (TnT), and tropomyosin. Cardiac muscle contraction is initially triggered by the binding of $Ca^{2+}$ to troponin C, which activates the thin filament via a series of intermolecular events involving TnI, TnT and tropomyosin, enabling ATP hydrolysis-driven movement of myosin cross-bridges to actin in thin filaments for force development and cell shortening [2,3].

On the other hand, cardiac myosin binding protein-C (MyBP-C) is a thick- filament-associated protein, and phosphorylation of MyBP-C by protein kinase A (PKA) and $Ca^{2+}$-calmodulin kinase 2 δ is a key determinant of the speed and force of cardiac contraction [4]. Myosin regulatory light chain (RLC) is also a thick-filament-associated protein and phosphorylation of RLC was recently demonstrated to increase force development as well as the $Ca^{2+}$ sensitivity of skinned cardiac fibers [5]. Importantly, altered RLC phosphorylation is known to contribute to compensatory responses and contractile dysfunction in human disease [6]. However, kinases that phosphorylate RLC in the heart have not been clearly identified, although there are three primary candidates. One is a cardiac myosin light chain kinase (cMLCK) regulated by the cardiac homeobox protein Nkx2.5 during development [7–9] and the others are a zipper-interacting protein kinase (ZIPK) [10,11] and a protein kinase C (PKC) [12,13]. The extent of RLC phosphorylation is determined by the balanced activities of kinases that phosphorylate RLC and myosin phosphatases, the regulatory mechanisms of which are now emerging [14].

The role of the cyclic AMP (cAMP) pathway in RLC phosphorylation in the heart also remains unclear, although β-adrenergic receptor (β-AR) stimulation was reported to increase RLC phosphorylation [15,16]. Recently, exchange protein activated by cAMP (Epac) was identified as a new PKA-independent sensor. Epac has two isoforms (Epac1 and Epac2). Epac1 is ubiquitously expressed, including heart, and has a single cAMP-binding site, whereas Epac2 contains a second cAMP-binding site and is mainly localized to brain and endocrine tissue [17,18]. However, the involvement of Epac1 in the $Ca^{2+}$ sensitivity of skinned myocardium has not been established.

Here, we examined the influence of Epac1 on the phosphorylation of myofilament-associated proteins, and its effects on the contractility of cardiac myofilaments, $Ca^{2+}$ sensitivity of force and ATPase activity in skinned myocardium, and its role in the pathogenesis of heart failure by means of a series of studies in transgenic mice with cardiac-specific overexpression of Epac1 (Epac1TG) as well as in wild-type (WT)

control mice by activating endogenous Epac1 with 8-(4-chlorophenylthio)-2'-O-Me-cAMP-AM (8CPT-AM), an Epac-specific cAMP activator.

## Materials and methods

### Ethical approval

All animal experiments complied with the ARRIVE guidelines [19] and were carried out in accordance with the National Institutes of Health guide for the care and use of laboratory animals [20] and institutional guidelines. The experimental protocol was approved by the Animal Care and Use Committee of Tsurumi University (No. 29A041) and Yokohama City University (F-A-21-067).

### Mice

Transgenic mice with cardiac-specific overexpression of Epac1 on a C57BL/6J background were developed using human Epac1 cDNA with the mouse α-myosin heavy chain promoter (Epac1TG) as described previously [21]. All experiments were performed in 4- to 6-month-old male Epac1TG and non-transgenic control mice (NTG) or male WT mice on a C57BL/6J background obtained from CLEA Japan (Tokyo, Japan) to examine the role of endogenous Epac1. Mice were group-housed (approximately 3 mice per cage) at 23°C under a 12-12 light/dark cycle with lights on at 8:00 AM. Both food and water were available ad libitum. After tissue extraction, the mice were killed by cervical dislocation.

### Skinned myocardial preparations

Skinned fiber bundles were prepared from LV papillary muscles as described previously with slight modifications [22–24]. Mice were anaesthetized with sodium pentobarbital (50 mg/kg, i.p.), and hearts were rapidly excised. Oxygenated Tyrode's solution was immediately infused into the coronary arteries via the aorta to remove blood from the myocardium. LV papillary muscles were dissected into fiber bundles (1.5–2.0 mm long and 150−200 μm in diameter) in the ice-cold oxygenated Tyrode's solution, and the fiber bundles were then fixed isometrically and incubated in a chemically skinning solution containing (mM): EGTA 5; Mg-ATP 2; free ATP 5; Na-propionate 122; Na-azide 10; imidazole 6 (pH 7.0) and 1% (v/v) Triton X-100 for 2 h at 4°C. After the skinning procedure, the demembranated muscle preparations were stored in an equivolume mixture of glycerol and the skinning solution without Triton X-100 at −20°C. They were used for experiments within 2 days. All solutions also contained 0.3% (v/v) protease inhibitor cocktail (P8340, Sigma-Aldrich, St. Louis, MO).

### Experimental solutions and apparatus

The relaxing solution (pCa 8.0) and the activating solutions with various $Ca^{2+}$ concentrations (pCa 6.1, 5.8, 5.5 and 4.6) were prepared as described previously by us [22–24]. The preactivating solution had the same composition as the relaxing solution, except that the EGTA concentration was reduced to 0.1 mM. The experimental apparatus was essentially the same as described previously [23,24]. The skinned preparation was mounted horizontally in an experimental chamber (volume 45 mL) between stainless steel hooks with fast-setting glue (collodion) via a loop of silk strand at either end of the preparation. One of the stainless steel hooks was connected to a force transducer (AE 801, SensoNor, Horten, Norway) mounted on a micromanipulator (Narishige Scientific Institute Laboratory, Tokyo, Japan) that could be moved to adjust the preparation length. The experimental chamber had quartz windows to allow transmission of near-ultraviolet light for measurement of nicotinamide adenine dinucleotide (NADH) absorbance as an index of ATPase activity, as in the previous studies [23,24]. The NADH absorbance signal was measured with a dual-wavelength spectrophotometer (JASCO Co., Tokyo, Japan). The force and NADH absorbance were simultaneously recorded with an A/D converter (Powerlab/8SP, ADInstruments, New South Wales, Australia) connected to a computer. The solution in the chamber was constantly stirred with a small magnet during the experiments. The

temperature of the solutions was kept at 22 °C by circulating temperature controlled water through a brass block beneath the experimental chamber. Sarcomere length of the preparation was measured by means of a light diffraction method with a He-Ne laser (model GLG5350, NEC, Tokyo, Japan).

## Measurements of isometric force and ATPase activity

The skinned preparation was first equilibrated for 10 min in the relaxing solution and the sarcomere length was adjusted to 2.2 μm, at which the preparation length and diameter were measured with a binocular micrometer. The preparation was activated twice at saturating [$Ca^{2+}$] (pCa 4.6) to confirm the amount of force deterioration. If the isometric force of the second activation was greater than 90% of the first and the resting tension returned to the initial level, the preparation was retained for the current experiments. It was then activated to contact isometrically at various pCa values ranging from 6.1 to 4.6. The solution was changed after both the isometric force and the rate of the light extinction corresponding to the NADH consumption reached steady-state values. The ATPase activity was measured by means of the method utilizing two enzymatic reactions [23,24]. Briefly, one reaction is ADP + phosphoenol pyruvate → ATP + pyruvate, catalyzed by pyruvate kinase and the other reaction is pyruvate + NADH → lactate + NAD, catalyzed by LDH. The ATPase activity associated with the $Ca^{2+}$-activated force development was obtained by subtracting the NADH absorbance change of the resting preparation from that of the $Ca^{2+}$-activated preparation. The values of force and ATPase activity were normalized to the cross-sectional area and volume of the skinned preparations, respectively. To determine the $Ca^{2+}$ sensitivity (pCa$_{50}$), the relative pCa-force and pCa-ATPase activity relations were fitted to the Hill equation: $P = [Ca^{2+}]^n / ([Ca^{2+}]^n + [Ca_{50}]^n)$ where $P$ is the parameter of interest (isometric force or ATPase activity), [$Ca_{50}$] is the $Ca^{2+}$ concentration required for half-maximal activation (pCa$_{50}$ = -log[$Ca_{50}$]) and $n$ represents the slope of the relationship (Hill coefficient), as described previously [23,24].

## Determination of phosphorylation status of myofibrillar proteins

Phosphorylation status of myofilament proteins were determined as previously described [22]. Myofilament proteins were separated on 10–20% gradient polyacrylamide gels (E-T1020L, ATTO, Tokyo, Japan). Phosphorylated proteins were detected by using Pro-Q Diamond Stain as suggested by the manufacture (Thermo Fisher Scientific, Waltham, MA). Subsequently, the gels were stained with SYPRO Ruby (Thermo Fisher Scientific) to visualize total proteins. The stained gels were scanned using a laser-scanning instrument (FLA-3000G, Fuji Photo Film, Tokyo, Japan). The fluorescence intensities of the bands for RLC, TnI, TnT and MyBP-C were evaluated with a densitometer (Science Lab 99 Image Gauge; Fuji Photo Film), and the ratio of Pro-Q Diamond dye to SYPRO Ruby dye signal intensities (Pro-Q/SYPRO) for each band was calculated to normalize the phosphorylation level to the total amount of protein.

## Pharmacological activation of Epac or PKA

To determine the effects of pharmacological activation of Epac on the phosphorylation status and the function of cardiac myofilaments, the skinned preparations were isometrically fixed and incubated with 8CPT-AM, BIOLOG Life Science Institute, Bremen, Germany) (1, 5 or 50 μM), an Epac-specific activator, or 6-Bnz-cAMP (BIOLOG Life Science Institute) (100 μM), a PKA-specific activator, in the relaxing solution for 30 min at 22°C. The doses of 8CPT-AM and 6-Bnz-cAMP were as previously described [25–28]. In addition, to elucidate Epac-mediated signaling pathways involved in the regulation of myofilament function, the skinned preparations were also incubated with 8CPT-AM (5 μM) in the presence of the inhibitors to the potential downstream molecules. The inhibitors used in this study were as follows: an inhibitor of multiple phospholipase C (PLC) isoforms, U73122 (Calbiochem) (5 μM); a broad specificity PKC inhibitor, bisindolylmaleimide-1 (BIM) (Calbiochem) (1 μM); a calcium/calmodulin-dependent protein kinase type II (CaMKII) inhibitor, KN-93 (Sigma-Aldrich) (2 μM); a MLCK inhibitor, ML-7 (Calbiochem) (10 μM); a ZIPK inhibitor, HS38 (Sigma-Aldrich) (50 μM). ML-7 was shown to work as a cMLCK inhibitor in previous studies [28–30]

The doses of the inhibitors have been shown to be effective in modulating biological activities [25–28,31].

## Western blotting

Western blotting was conducted using commercially available antibodies [32] as follows: primary antibodies against myosin phosphatase target subunit 1 (MYPT1) (#612165, BD Transduction Laboratories, 1:1000 dilution), MYPT1/2 (#ab32519, Abcam, 1:2000 dilution), phospho-MYPT (Thr696, #ABS45, Upstate Biotechnology, 1:2000 dilution), Epac1 (#4155, Cell Signaling Technology, 1:1000 dilution), PKAα catalytic subunit (#sc-903, Santa Cruz Biotechnology, 1:200 dilution), PLCε (#ab121476, Abcam, 1:500 dilution), PKC (#12919-1-AP, Proteintech, 1:2000 dilution), CaMKII (#4436, Cell Signaling Technology, 1:1000 dilution), cMLCK (#21527-1-AP, Proteintech, 1:1000 dilution), ZIPK (ab210528, Abcam, 1:1000 dilution), histone H3 (#4499, Cell Signaling Technology, 1:2000 dilution), GAPDH (#sc-25778, Santa Cruz Biotechnology, 1:200 dilution), α-actinin (#A7811, Sigma-Aldrich, 1:2500 dilution), phospho-RLC (Ser15, #PA5-104265, Invitrogen, 1:500 dilution), RLC (#10906-1-AP, Proteintech, 1:3000 dilution). Horseradish peroxidase-conjugated anti-rabbit IgG (NA934, GE Healthcare) or anti-mouse IgG (NA931, GE Healthcare) was used as a secondary antibody. The primary and secondary antibodies were diluted in Can Get Signal Immunoreaction Enhancer Solution (Toyobo, Osaka, Japan). The blots were visualized with enhanced chemiluminescence solution (ECL PrimeWestern Blotting Detection Reagent, GE Healthcare) and scanned with a densitometer (Amersham Imager 600, GE Healthcare).

## MHC composition

Myosin heavy chain (MHC) isoform composition in left ventricular muscles was analyzed by means of SDS-PAGE, followed by silver staining (Silver Staining Kit, GE Healthcare) as previously described [33,34]. The stained bands were scanned and analyzed with a densitometer (Amersham Imager 600, GE Healthcare). The bands of MHC isoforms (i.e., MHC-α or β) were identified by using neonatal ventricular samples containing both α and β isoforms [35,36].

## Physiological experiments

Mice were anesthetized with isoflurane vapor titrated to maintain the lightest anesthesia possible and echocardiographic measurements were performed as described previously [37]. Long-term infusion of isoproterenol (ISO) was performed for 7 days at 60 mg/kg/day and control mice received an identical dose of saline (Vehicle) as described previously by us and others [37–39].

## Isolated heart experiments

To determine the effects of pharmacological activation of Epac on intact myocardium, 8CPT-AM (BIOLOG Life Science Institute), an Epac-specific activator, was dissolved in a Langendorff perfusion model. Mice were treated with heparin (500 U/kg, i.p.) and anesthetized with sodium pentobarbital (50 mg/kg, i.p.). The hearts were quickly removed, mounted on the aortic cannula of a Langendorff perfusion system and perfused with an oxygenated Tyrode's solution (mM): NaCl 130; $CaCl_2$ 2; KCl 4; $MgCl_2$ 1; $NaH_2PO_4$ 0.4; $NaHCO_3$ 10; glucose 5.5 (pH 7.4) in the presence or absence of 8CPT-AM (2 μM) for 30 min at 37°C [40,41]. To exclude the involvements of PKA activation, the hearts were also perfused with 8CPT-AM (2 μM) in the presence of H-89 (Sigma-Aldrich) (5 μM), a highly specific inhibitor of PKA [42]. After the completion of each treatment, the left ventricles were frozen in liquid nitrogen, and stored at −80ºC for later analysis.

## Statistical analysis

Data are expressed as means ± SD. Open circles in the bar charts show individual data from biological replicates, each with 3 technical replicates that were averaged for subsequent analysis.

Comparison of data was performed using a Student's unpaired *t*-test when compared between 2 groups (Figs 2B,6C–6G,7B,7D and S2B, S2C, S4B, S8C–S8G Figs in S2 Data) or one-way ANOVA followed by Tukey's *post hoc* test for 3 or more groups (Figs 1B,1C,3B,4B,5B,5C,5E,5F,7A,7C,7E and S1B–S1G, S3B, S4C, S5, S6B, S6C Figs in S2 Data). The criterion of significance was taken as *P*<0.05.

## Results

### Epac1 is overexpressed in skinned myocardium prepared from Epac1TG

In order to examine the role of Epac1 in the response of cardiac myofilaments to β-AR stimulation, we first examined its expression in total myocardium homogenate or skinned myocardium prepared from Epac1TG and non-transgenic mice (NTG). Epac1 expression was observed in total myocardium homogenate prepared from NTG and Epac1TG (**Fig 1A, left**) and the magnitude of the increase in Epac1TG was approximately 16-fold, compared to that in NTG (*P* < 0.01 by one-way ANOVA) (**Fig 1B, left**). More importantly, Epac1 expression was also observed in skinned myocardium prepared from NTG (*n* = 6) and Epac1TG (*n* = 6) (**Fig 1A, right**) and the magnitude of the increase was approximately 16-fold, compared to NTG (*P* < 0.01 by one-way ANOVA) (**Fig 1B, right**).

Epac1 is expressed not only in the cytoplasm, but also in the nuclear region in NTG and Epac1TG [21], which is consistent with previous findings in cultured cells [43,44]. We thus examined the expression of histone H3 and confirmed that it is equally expressed in total and skinned myocardium in both NTG and Epac1TG (S1 Fig of S1 Data). This result suggests that Epac1 expressed in the nuclear region is present in both intact ventricle and skinned myocardium (**Fig 1A**).

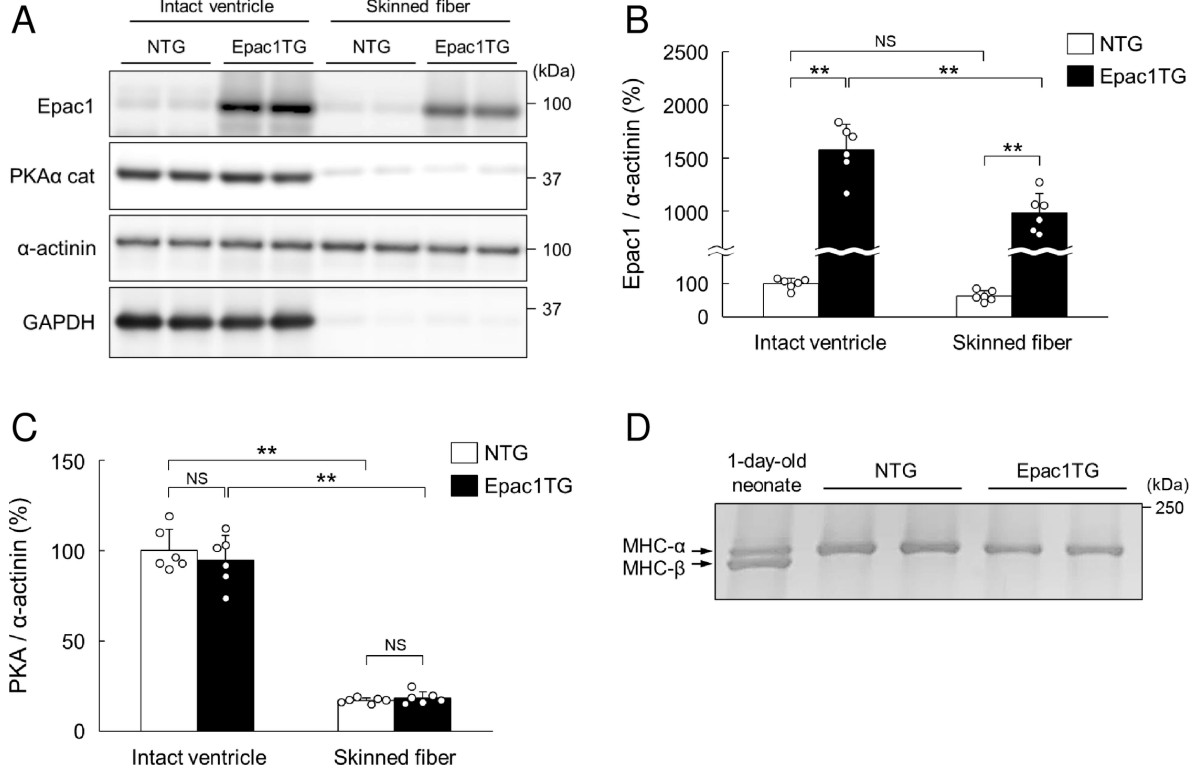

**Fig 1. Epac1 expression in the hearts of NTG and Epac1TG.**

Conversely, the expression level of PKA was similar between NTG and Epac1TG in total (NTG vs. Epac1TG: 100 ± 12 vs. 95 ± 14, $P$ = NS by one-way ANOVA, $n$ = 6 each) and in skinned myocardium (NTG vs. Epac1TG: 17 ± 1 vs. 18 ± 3, $P$ = NS by one way ANOVA, $n$ = 6 each) (**Fig 1C**). This result suggests that small amounts of PKA might similarly co-localize with cardiac myofibrils in both NTG and Epac1TG, in contrast to the case of Epac1.

## No isoform shift of MHC is observed in Epac1TG

We have previously demonstrated that left ventricle/tibial length ratio was slightly but significantly increased (approximately 16%) in Epac1TG, compared to that in NTG [21]. Neonatal isoforms of sarcomeric proteins such as MHC-β are known to be re-expressed in the adult heart during pathological hypertrophy or in response to pathophysiological stimuli [45], which might alter $Ca^{2+}$ sensitivity, force generation and ATPase consumption [46]. We thus examined the expression of MHC (**Fig 1D**) in total myocardium. However, expression of MHC-β was not observed in total myocardium prepared from either NTG or Epac1TG (**Fig 1D**).

These data suggest that we can rule out the potential influence of re-expression of neonatal isoforms of sarcomeric protein, MHC-β, during our examination of the role of Epac1 in $Ca^{2+}$-sensitivity, force generation and ATPase consumption in the heart of Epac1TG.

## RLC phosphorylation is greater in skinned myocardium of Epac1TG

We next examined the phosphorylation status of cardiac myofibril components, RLC, TnI, TnT and MyBP-C, by SDS-PAGE analysis (**Fig 2A**), because their activation state is closely associated with Ca2+ sensitivity [47].

We first validated Pro-Q Diamond Stain for quantification. We prepared myocardium from hearts treated with (negative control) or without (positive control) Langendorff perfusion with calcium-free Tyrode's solution for 1 hr, and examined the

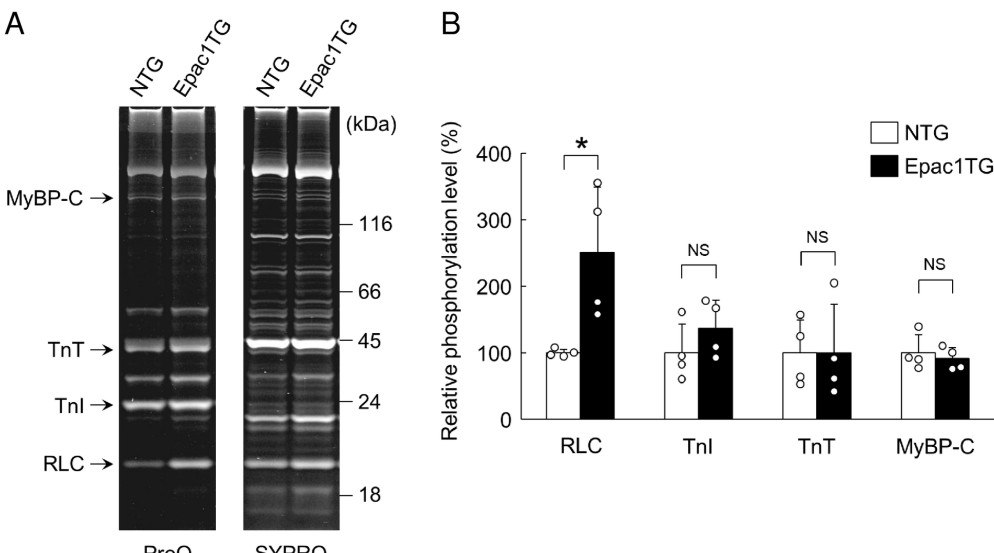

**Fig 2. Phosphorylation status of myofilament proteins prepared from Epac1TG.** (A) Representative SDS-PAGE patterns (10-20% gradient gel) of skinned myocardium prepared from NTG (left lane) and Epac1TG (right lane). The gel was stained with Pro-Q Diamond (specific for phosphorylated proteins) and subsequently stained with SYPRO Ruby (for total proteins). RLC, myosin regulatory light chain; TnI, troponin I; TnT, troponin T; MyBP-C, myosin binding protein-C. (B) RLC phosphorylation in Epac1TG was significantly greater than that in NTG (*$P$<0.05 by unpaired $t$-test). However, phosphorylation levels of TnI, TnT and MyBP-C were similar between the two groups ($P$=NS by unpaired $t$-test). The mean phosphorylation level in NTG was taken as 100% in each determination. Bar blots represent means±SD and open circles show individual data from biological replicates of NTG and Epac1TG ($n$=4 each), each with 3 technical replicates.

RLC phosphorylation status by Pro-Q/SYPRO staining and western blotting (S2A Fig of S1 Data). The RLC phosphorylation status was significantly increased by approximately two-fold as determined by both Pro-Q/SYPRO staining (negative vs. positive: 100 ± 29 vs. 211 ± 27%, *P* < 0.001 by unpaired *t*-test) and western blotting (negative vs. positive: 100 ± 55 vs. 237 ± 35%, *P* < 0.001 by unpaired *t*-test) (S2B and S2C Fig of S1 Data), indicating that quantification of phosphorylation levels in myofilaments by means of Pro-Q/SYPRO staining gives similar results to western blotting, as shown previously [48]

RLC phosphorylation in Epac1TG was significantly greater than in NTG (NTG vs. Epac1TG: 100 ± 6 vs. 251 ± 98%, *P* < 0.05 by unpaired *t*-test) (Fig 2B). However, phosphorylation levels of TnI, TnT and MyBP-C were similar in NTG and Epac1TG (*P* = NS by unpaired *t*-test) (Fig 2B).

These data suggest that Epac1 might increase RLC phosphorylation via cAMP/Epac1 signaling, thereby causing increased Ca2+ sensitivity of force and ATPase activity, with increased tension cost in skinned myocardium from Epac1TG, compared to NTG [49].

**Epac activation with 8CPT-AM increases RLC phosphorylation in the isolated heart perfused with the Langendorff method**

We also examined the phosphorylation status of cardiac myofibril components, RLC, TnI, TnT and MyBP-C, by SDS-PAGE analysis in total cardiac homogenate prepared from wild-type (WT) hearts perfused with the Langendorff method (Fig 3A). Hearts were perfused with an oxygenated Tyrode's solution with or without 8CPT-AM (2 μM), an Epac-specific activator, or 8CPT-AM (2 μM) plus H-89 (5 μM), a PKA-specific inhibitor, for 30 min, and we confirmed that RLC phosphorylation was significantly increased in the heart perfused with Tyrode's solution containing 8CPT-AM (2 μM) (Control

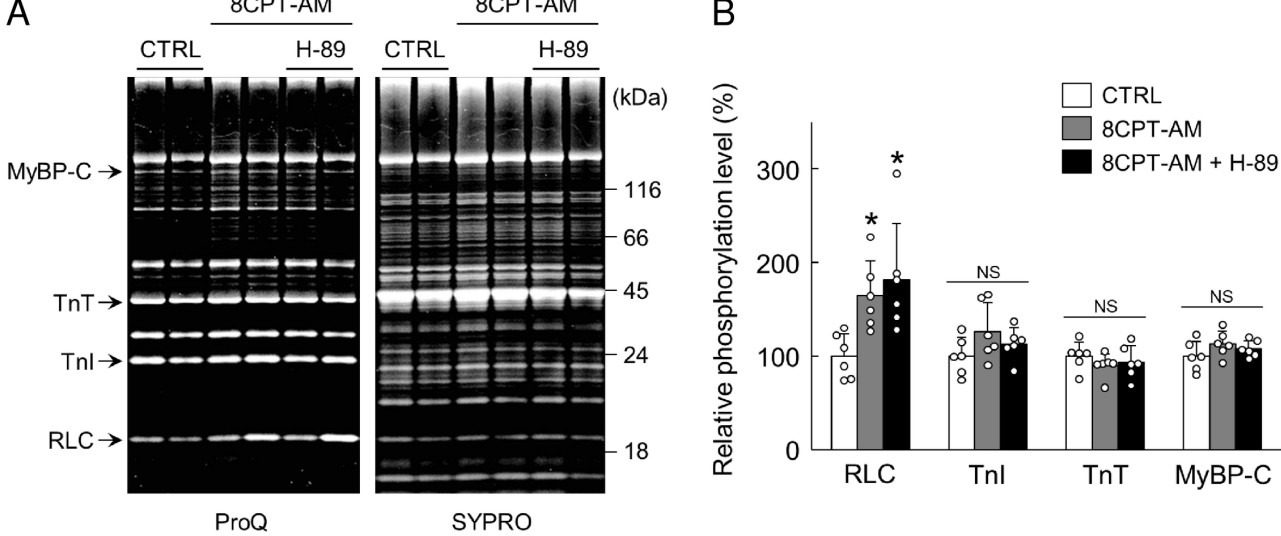

**Fig 3. Epac activation with 8CPT-AM induced RLC phosphorylation in the perfused hearts.** (A) Representative phosphorylation status of cardiac myofibril components, RLC, TnI, TnT and MyBP-C, as determined by SDS-PAGE analysis of WT hearts perfused with or without Epac activator, 8CPT-AM (2 μM), or 8CPT-AM (2 μM) plus PKA inhibitor, H-89 (5 μM), for 30 min. The gel was stained first with Pro-Q Diamond and then stained with SYPRO Ruby. RLC, myosin regulatory light chain; TnI, troponin I; TnT, troponin T; MyBP-C, myosin binding protein-C. (B) RLC phosphorylation was significantly increased in the presence of 8CPT-AM (2 μM) (*P* < 0.05 vs. Control by one-way ANOVA, *n* = 6 each) or 8CPT-AM (2 μM) plus H-89 (5 μm) (*P* < 0.05 vs. Control by one-way ANOVA, *n* = 6 each), but the magnitudes of the increase were similar (*P* = NS by one-way ANOVA, *n* = 6 each). Phosphorylation levels of TnI, TnT and MyBP-C were similar among the three groups (*P* = NS by one-way ANOVA, *n* = 6 each). The mean phosphorylation level in the control was taken as 100% in each case. Bar blots represent means ± SD and open circles show individual data from biological replicates of Control, 8CPT-AM, 8CPT-AM + H89 (*n* = 6 each), each with 3 technical replicates.

vs. 8CPT-AM; 100 ± 24 vs. 165 ± 37%, P < 0.05 by one-way ANOVA) and 8CPT-AM (2 µM) plus H-89 (5 µM) (Control vs. 8CPT-AM + H89; 100 ± 24 vs. 182 ± 60%, P < 0.05 by one-way ANOVA) (**Fig 3B**). However, phosphorylation levels of TnI, TnT and MyBP-C were similar among the three groups (**Fig 3B**).

Basal phosphorylation levels of PKA targets such as MyBP-C and TnI were not altered by H89, in accordance with the previous reports by other groups, probably due to the low levels of PKA-dependent phosphorylation [25,50].

These data indicate that endogenous Epac1 specifically increased RLC phosphorylation, independently of PKA.

## Epac activation with 8CPT-AM increases RLC phosphorylation in skinned myocardium prepared from wild-type mice

We examined the effects of endogenous Epac1 activation with 8CPT-AM (1 µM, 5 µM and 50 µM for 30 min, respectively) on the phosphorylation status of cardiac myofibril components, RLC, TnI, TnT and MyBP-C, by SDS-PAGE analysis (S3A Fig of S1 Data) and found that RLC phosphorylation in skinned myocardium prepared from wild-type (WT) mice and treated with 8CPT-AM at 5 µM and 50 µM was significantly greater than in the control (Control vs. 8CPT-AM [5 µM]: 100 ± 17 vs. 159 ± 16%, P < 0.05 by one-way ANOVA, Control vs. 8CPT-AM [50 µM]: 100 ± 17 vs. 165 ± 31%, P < 0.05 by one-way ANOVA (S3B Fig of S1 Data). However, phosphorylation levels of TnI, TnT and MyBP-C in the cardiac myofibril proteins were similar with or without 8CPT-AM treatment (P = NS by one-way ANOVA) (S3B Fig of S1 Data).

We also treated the skinned myocardium with 6-Bnz-cAMP (100 µM for 30 min), a PKA-specific activator on the phosphorylation status of cardiac myofibril components, RLC, TnI, TnT and MyBP-C, and no significant increases of the phosphorylation levels were observed among them (S3B Fig of S1 Data), probably due to the small amount of PKA in skinned myocardium (**Fig 1A** and **1C**).

We next examined the effects of Epac activation with 8CPT-AM (1 µM and 5 µM for 30 min, respectively) on phosphorylation status of skinned myocardium prepared from Epac1TG and NTG (S4A Fig of S1 Data). RLC phosphorylation was significantly greater in Epac1TG than in NTG at baseline (NTG [Control] vs. Epac1TG [Control]: 100 ± 9 vs. 136 ± 15%, P < 0.05 by unpaired t-test) and also after the treatment of 8CPT-AM at 1 µM (NTG vs. Epac1TG; 98 ± 16 vs. 160 ± 13%, P < 0.01 by unpaired t-test) (S4B Fig of S1 Data). However, RLC phosphorylation level was similar and might be saturated in both NTG and Epac1TG after the treatment of 8CPT-AM at 5 µM (NTG vs. Epac1TG; 143 ± 32 vs. 154 ± 16%, P = NS by unpaired t-test) (S4B Fig of S1 Data). However, phosphorylation levels of TnI, TnT and MyBP-C were similar at baseline and after the treatment of 8CPT-AM (S4C Fig of S1 Data).

These data indicate that activation of endogenous Epac1 and overexpression of Epac1 might increase RLC phosphorylation via cAMP/Epac1 signaling in cardiac myofilament proteins without altering phosphorylation levels of TnI, TnT and MyBP-C.

## Epac activation induces RLC phosphorylation via PLC/PKC

Three kinases that phosphorylate RLC in the heart have been reported: one is a $Ca^{2+}$/calmodulin-dependent cMLCK [8,9,51] and the others are a ZIPK [10,11] and a PKC [12,13].

We and the other groups have demonstrated that Epac activation might induce cardiac remodeling and dysfunction via the PLC/PKC pathway [27,37]. We thus examined the effects of Epac activation with 8CPT-AM (5 µM for 30 min) on RLC phosphorylation with or without the presence of PLC inhibitor (U73122: 5 µM), PKC inhibitor (BIM: 1 µM), CaMKII inhibitor (KN-93: 2 µM), MLCK inhibitor (ML-7: 10 µM) or ZIPK inhibitor (HS38: 50 µM) in skinned myocardium (**Fig 4A**). Epac activation with 8CPT-AM significantly increased RLC phosphorylation, compared to that in the control (Control vs. 8CPT-AM; 100 ± 12 vs. 161 ± 21%, P < 0.05) (**Fig 4B**). Pharmacological inhibition of CaMKII (Control vs. KN-93; 100 ± 12 vs. 159 ± 18%, P < 0.05), MLCK (Control vs. ML-7; 100 ± 12 vs. 174 ± 23%, P < 0.01) or ZIPK (Control vs. HS38; 100 ± 12 vs. 172 ± 19%, P < 0.01) did not alter the increase of RLC phosphorylation in response to 8CPT-AM, but pharmacological inhibition of PLC (Control vs. U73122; 100 ± 12 vs. 114 ± 13%, P = NS) or PKC (Control vs. BIM; 100 ± 12 vs. 128 ± 21%, P =

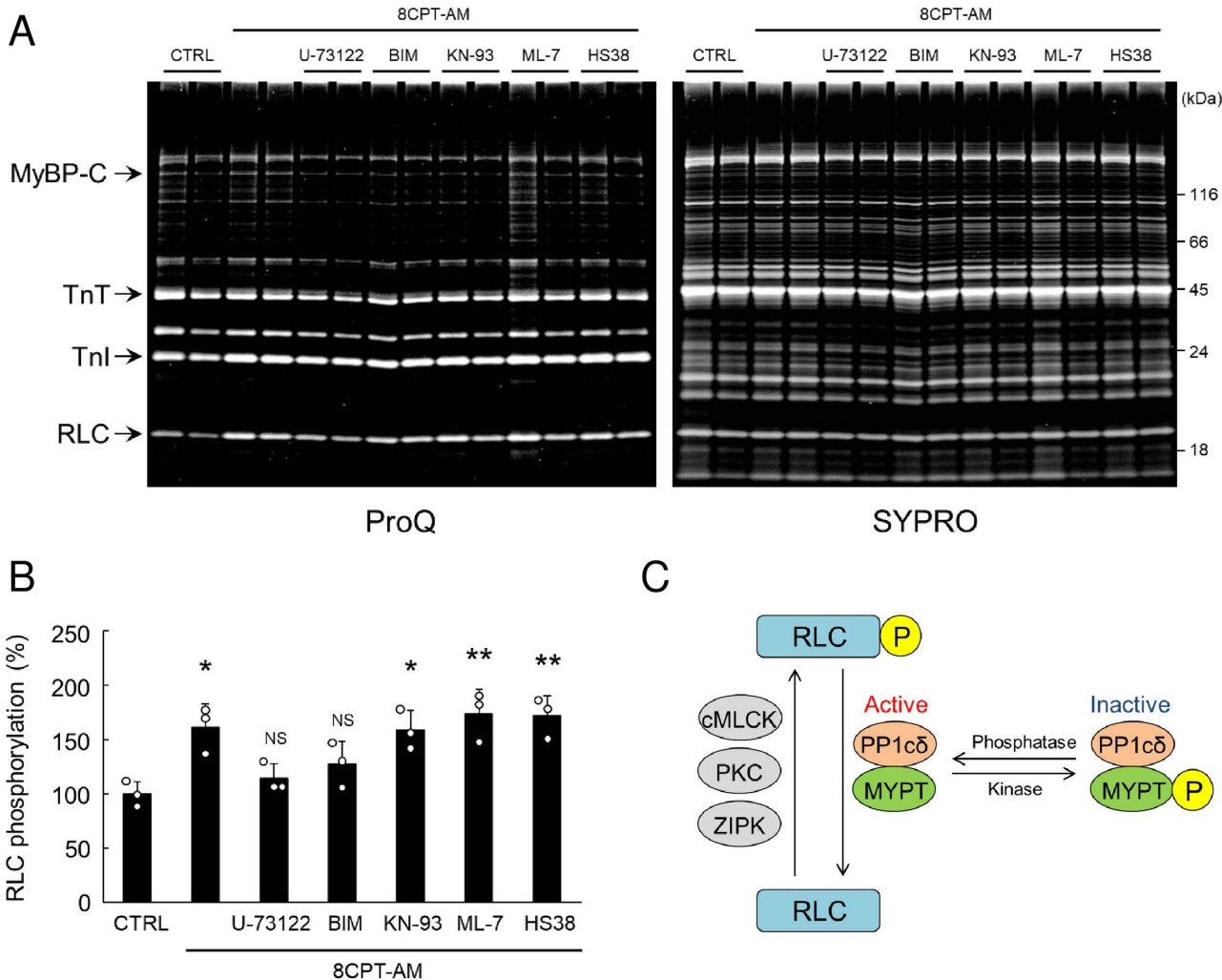

**Fig 4. Epac activation increased RLC phosphorylation via PLC/PKC.** (A) Representative SDS-PAGE patterns of skinned myocardium prepared from WT and treated with 8CPT-AM (5 μM for 30 min) in the presence or absence of PLC inhibitor (U73122, 5 μM), PKC inhibitor (BIM, 1 μM), CaMKII inhibitor (KN-93, 2 μM), MLCK inhibitor (ML-7, 10 μM) or ZIPK inhibitor (HS38, 50 μM). The gel was stained with Pro-Q Diamond and subsequently stained with SYPRO Ruby. (B) Pharmacological inhibition of CaMKII, MLCK or ZIPK did not alter the increase of RLC phosphorylation in response to 8CPT-AM (*$P < 0.05$, **$P < 0.01$ vs. Control by one-way ANOVA), but pharmacological inhibition of PLC or PKC attenuated Epac-mediated RLC phosphorylation ($P$ = NS vs. Control by one-way ANOVA). The mean phosphorylation level in the control was taken as 100% in each determination. (C) Scheme illustration of phosphorylation and dephosphorylation of RLC. Myosin phosphatase is composed of a 37-kDa catalytic subunit (PP1cδ) that is responsible for dephosphorylation of its highly specific substrate (RLC) and 130-kDa regulatory myosin binding subunit (MYPT). Phosphorylation of MYPT by Rho kinase results in loss of myosin phosphatase activity. Bar blots represent means ± SD and open circles show individual data from biological replicates ($n$ = 3 each), each with 3 technical replicates.

NS) inhibited Epac-mediated RLC phosphorylation (**Fig 4B**). However, Epac activation did not alter phosphorylation levels of TnI, TnT and MyBP-C with or without inhibitors of PLC, PKC, CaMKII, MLCK or ZIPK ($P$ = NS by one-way ANOVA) (S5 Fig of S1 Data).

We also examined the expression of PLC, PKC, CaMKII, cardiac MLCK (cMLCK) and ZIPK in total myocardium homogenate and skinned myocardium prepared from Epac1TG and NTG, and confirmed that they were equally expressed in myocardium prepared from Epac1TG and NTG (S1 Fig of S1 Data).

These data suggest that endogenous Epac1 activation with 8CPT-AM specifically increased RLC phosphorylation via PLC/PKC.

## MYPT phosphorylation is increased in Epac1TG

The ratio of myosin light chain phosphatase to kinase determines the level of RLC phosphorylation [52]. Myosin phosphatase is composed of a 37-kDa catalytic subunit (PP1c$\delta$), which is responsible for highly specific dephosphorylation of its substrate RLC, and 130-kDa regulatory myosin binding subunit (myosin phosphatase targeting protein; MYPT) (Fig 4C). Two isoforms of MYPT (MYPT1 and MYPT2) are expressed in the heart: MYPT1 accounts for about one-third and MYPT2 accounts for about two-thirds [53]. Rho-kinase-mediated phosphorylation of MYPT1 (Thr 696) and MYPT2 (Thr 646) results in loss of myosin phosphatase activity [54] (note that Thr 646 is equivalent to Thr 696 in MYPT1 and is similarly detected by anti-phospho-MYPT1 (Thr 696) antibody). MYPT2 was recently demonstrated to be the major regulatory protein for cardiac MLCK [53].

We thus examined the effects of Epac activation with 8CPT-AM (1 μM or 5 μM for 30 min) on phosphorylation of MYPT1 (Thr 696) and MYPT2 (Thr 646) in skinned myocardium prepared from NTG or Epac1TG (Fig 5A). In NTG, MYPT1 phosphorylation (Thr 696) and MYPT2 phosphorylation (Thr 646) levels were similar to the control after 8CPT-AM treatment at 1 μM (MYPT1: Control vs. 8CPT-AM [1 μM]; $100 \pm 11$ vs. $112 \pm 15\%$, $P = $ NS; MYPT2: Control vs. 8CPT-AM [1 μM]; $100 \pm 19$ vs. $97 \pm 28\%$, $P = $ NS), but they were significantly increased after 8CPT-AM treatment at 5 μM (MYPT1: $169 \pm 27\%$, $P < 0.05$ vs. Control; MYPT2: $192 \pm 30\%$, $P < 0.05$ vs. Control) (Fig 5B and 5C). In Epac1TG, basal MYPT1 phosphorylation (Thr 696) and MYPT2 phosphorylation (Thr 646) levels were significantly greater than in NTG (MYPT1: $159 \pm 28\%$, $P < 0.05$ vs. Control [NTG]; MYPT2: $187 \pm 34\%$, $P < 0.05$ vs. Control [NTG]), and might be saturated after 8CPT-AM treatment at 1 μM (MYPT1: $187 \pm 20\%$, $P < 0.01$ vs. Control [NTG]; MYPT2: $208 \pm 27\%$, $P < 0.05$ vs. Control [NTG]) and 5 μM (MYPT1: $188 \pm 23\%$, $P < 0.01$ vs. Control [NTG]; MYPT2: $201 \pm 44\%$, $P < 0.05$ vs. Control [NTG]) (Fig 5B and 5C).

These data suggest that activation of endogenous Epac1 with CPT-AM in WT and overexpression of Epac1 in Epac1TG increase the MYPT1 and MYPT2 phosphorylation levels.

## Epac-mediated MYPT phosphorylation is induced via PLC/PKC

We also examined the effects of Epac activation with 8CPT-AM (5 μM for 30 min) on MYPT1 (Thr 696) and MYPT2 (Thr 646) phosphorylation in the presence and absence of a PLC inhibitor (U73122; 5 μM), PKC inhibitor (BIM; 1 μM), CaMKII inhibitor (KN-93; 2 μM), MLCK inhibitor (ML-7; 10 μM) or ZIPK inhibitor (HS38; 50 μM) (Fig 5D). Epac activation with 8CPT-AM significantly increased MYPT1 phosphorylation (Thr 696) and MYPT2 phosphorylation (Thr 646), compared to the control (MYPT1: Control vs. 8CPT-AM; $100 \pm 19$ vs. $174 \pm 12\%$, $P < 0.05$ by one-way ANOVA; MYPT2: Control vs. 8CPT-AM; $100 \pm 20$ vs. $184 \pm 18\%$, $P < 0.05$ by one-way ANOVA) (Fig 5E and 5F). Pharmacological inhibition of CaMKII (Control vs. KN-93; MYPT1: $100 \pm 19$ vs. $165 \pm 20\%$; MYPT2: $100 \pm 20$ vs. $177 \pm 15\%$), MLCK (Control vs. ML-7; MYPT1: $100 \pm 19$ vs. $186 \pm 29\%$; MYPT2: $100 \pm 20$ vs. $206 \pm 28\%$) and ZIPK (Control vs. HS38; MYPT1: $100 \pm 19$ vs. $166 \pm 21\%$; MYPT2: $100 \pm 20$ vs. $183 \pm 41\%$) did not affect the increase of MYPT phosphorylation in response to 8CPT-AM treatment, but pharmacological inhibition of PLC (Control vs. U-73122; MYPT1: $100 \pm 19$ vs. $120 \pm 24\%$; MYPT2: $100 \pm 20$ vs. $127 \pm 33\%$) and PKC (Control vs. BIM; $100 \pm 19$ vs. $110 \pm 24\%$; MYPT2: $100 \pm 20$ vs. $125 \pm 19\%$) similarly inhibited Epac-mediated MYPT1 phosphorylation (Thr 696) and MYPT2 phosphorylation (Thr 646) (Fig 5E and 5F).

These data indicate that Epac-mediated MYPT1/2 phosphorylation might be mediated by PLC/PKC. More importantly, together with the data in Fig 4, the results suggest that Epac activation increased RLC phosphorylation via inhibition of myosin light chain phosphatase activity and stimulation of the kinase that phosphorylates RLC.

 

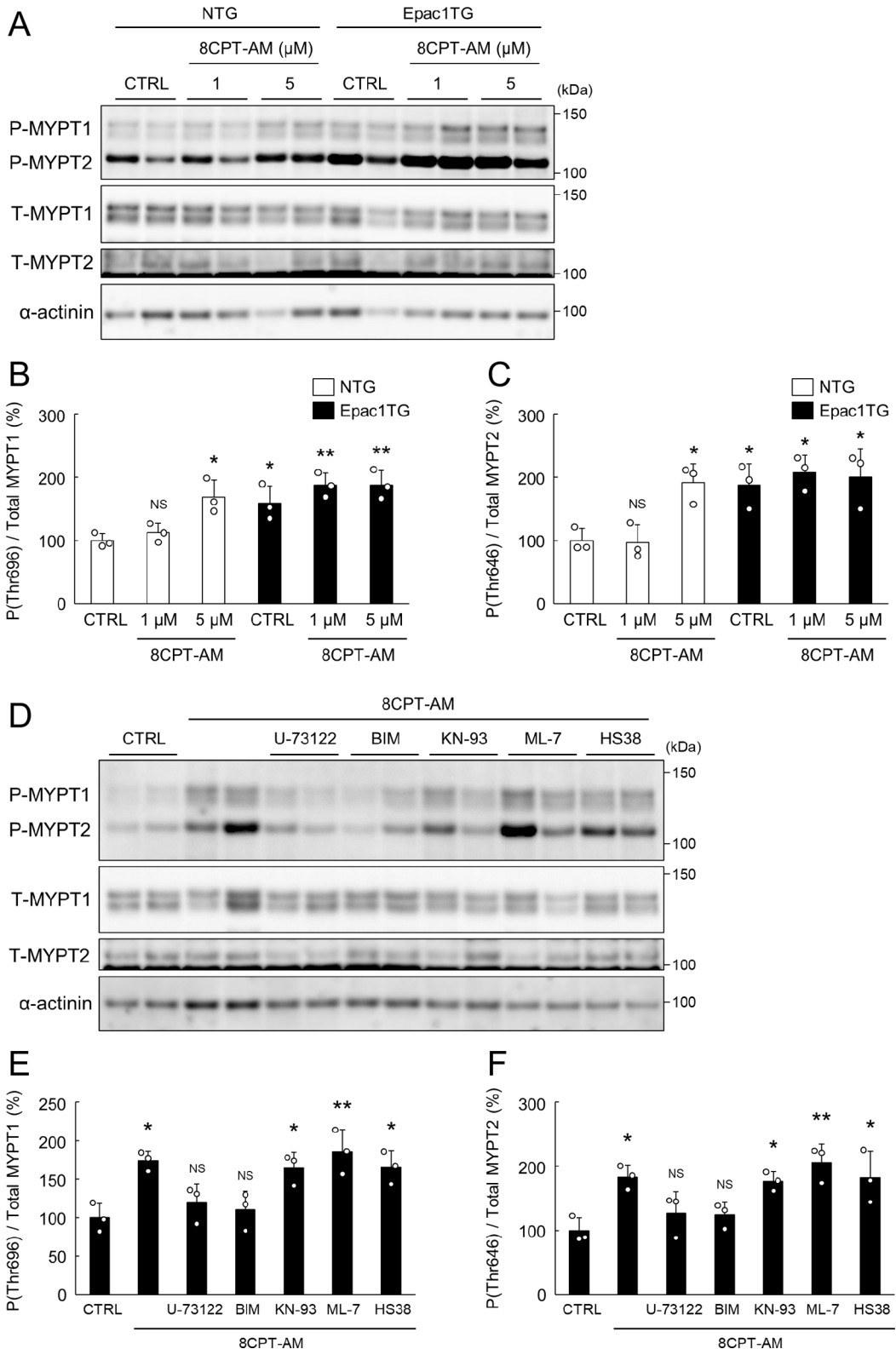

**Fig 5. Epac activation increased MYPT phosphorylation via PLC/PKC.** (A) Representative western blotting of phosphorylated (Thr 696) and total MYPT1 as well as phosphorylated (Thr 646) and total MYPT2 in skinned myocardium prepared from NTG and Epac1TG and treated with or without 8CPT-AM (1 µM or 5 µM for 30 min). (B, C) In NTG, phosphorylation of MYPT1 (Thr 696) **(B; left)** and MYPT2 (Thr 646) **(C; left)** were not altered by the

8CPT-AM treatment at 1 µM ($P$ = NS vs. Control (NTG) by one-way ANOVA), but those were significantly increased by the 8CPT-AM treatment at 5 µM (*$P$ < 0.05 vs. Control (NTG) by one-way ANOVA). In Epac1TG, phosphorylation of MYPT1 **(B; right)** and MYPT2 **(C; right)** might be saturated without 8CPT-AM and after 8CPT-AM treatment (1 µM and 5 µM) (*$P$ < 0.05, **$P$ < 0.01 vs. Control (NTG) by one-way ANOVA). The mean phosphorylation level in the control NTG was taken as 100% in each determination. (D) Representative western blotting of phosphorylated and total MYPT1/2 in skinned myocardium treated with 8CPT-AM (5 µM for 30 min) in the presence or absence of PLC inhibitor (U73122, 5 µM), PKC inhibitor (BIM, 1 µM), CaMKII inhibitor (KN-93, 2 µM), MLCK inhibitor (ML-7, 10 µM) or ZIPK inhibitor (HS38, 50 µM). (E, F) Pharmacological inhibition of CaMKII, MLCK or ZIPK did not alter the increase in the phosphorylation of MYPT1 **(E)** and MYPT2 **(F)** in response to the 8CPT-AM treatment (*$P$ < 0.05, **$P$ < 0.01 vs. Control by one-way ANOVA), but pharmacological inhibition of PLC or PKC attenuated Epac-mediated MYPT1/2 phosphorylation ($P$ = NS vs. Control by one-way ANOVA). The mean phosphorylation level in the control was taken as 100% in each determination. Bar blots represent means ± SD and open circles show individual data from biological replicates ($n$ = 3 each), each with 3 technical replicates.

### Epac activation with 8CPT-AM increases MYPT phosphorylation in isolated WT heart perfused with the Langendorff method

We also examined the phosphorylation status of MYPT1 (Thr 696) and MYPT2 (Thr 646) by western blotting in total myocardium homogenate prepared from WT hearts ($n$ = 6 for each treatment) perfused with the Langendorff method (S6A Fig of S1 Data). MYPT1 phosphorylation (Thr 696) and MYPT2 phosphorylation (Thr 646) were significantly increased in the heart perfused with Tyrode's solution containing 8CPT-AM (2 µM) (MYPT1: Control vs. 8CPT-AM; 100 ± 14 vs. 173 ± 27%, $P$ < 0.05 by one-way ANOVA; MYPT2: Control vs. 8CPT-AM; 100 ± 35 vs. 177 ± 56%, $P$ < 0.05 by one-way ANOVA) or 8CPT-AM plus H-89 (5 µM) (MYPT1: Control vs. 8CPT-AM + H89; 100 ± 14 vs. 187 ± 79%, $P$ < 0.05 by one-way ANOVA; MYPT2: Control vs. 8CPT-AM + H89; 100 ± 35 vs. 194 ± 58%, $P$ < 0.05 by one-way ANOVA) but the magnitude of the increase was similar ($P$ = NS) (S6B and S6C Fig of S1 Data).

These data, together with the data shown in **Fig 5**, indicate that activation of endogenous Epac specifically increased MYPT1 and MYPT2 phosphorylation, independently of PKA.

### Ca²⁺ sensitivity of force and ATPase activity are significantly greater in Epac1TG

As shown in **S7 Fig of** S1 Data, the $Ca^{2+}$-activated isometric force (lower traces in each panel) and ATPase activity, measured in terms of the consumption of NADH (upper traces in each panel) were recorded simultaneously in skinned myocardium prepared from NTG (S7 Fig of S1 Data, **upper**) and Epac1TG (S7 Fig of S1 Data, **lower**).

The average values of isometric force (**Fig 6A**) and ATPase activity (**Fig 6B**) in NTG and Epac1TG at the $Ca^{2+}$ concentration of 6.1, 5.8, 5.5, 5.1 and 4.6 (expressed as pCa = -log [$Ca^{2+}$]) were plotted. The curves representing the force-pCa relationship (**Fig 6A**) and the ATPase activity-pCa relationship (**Fig 6B**) each showed a leftward shift (i.e., an increase of $Ca^{2+}$ sensitivity) in Epac1TG, compared to those in NTG, suggesting that the $Ca^{2+}$ sensitivity of force and ATPase activity is significantly greater in Epac1TG than in NTG.

We also calculated the average of pCa producing 50% force or ATPase activity, i.e., $pCa_{50}$, and Hill coefficients by fitting the data of $Ca^{2+}$-activated force and $Ca^{2+}$-activated ATPase activity within the pCa range between 6.1 to 4.6 for Epac1TG and NTG (**Fig 6C and 6D**). The $pCa_{50}$ of isometric force was significantly greater in Epac1TG than in NTG (NTG vs. Epac1TG: 5.47 ± 0.04 vs. 5.60 ± 0.06, $P$ < 0.05 by unpaired $t$-test) (**Fig 6C, left**), with no change of the Hill coefficient (NTG vs. Epac1TG: 2.77 ± 0.17 vs. 2.88 ± 0.22, $P$ = NS by unpaired $t$-test) (**Fig 6D, left**). Similarly, the $pCa_{50}$ of ATPase activity was also significantly greater in Epac1TG than in NTG (NTG vs. Epac1TG: 5.52 ± 0.06 vs. 5.64 ± 0.04, $P$ < 0.05 by unpaired $t$-test) (**Fig 6C, right**), and there was no change of the Hill coefficient (NTG vs. Epac1TG: 2.72 ± 0.25 vs. 2.64 ± 0.24, $P$ = NS by unpaired $t$-test) (**Fig 6D, right**).

These data all support the idea that the $Ca^{2+}$ sensitivity of force and ATPase activity is significantly greater in Epac1TG than in NTG.

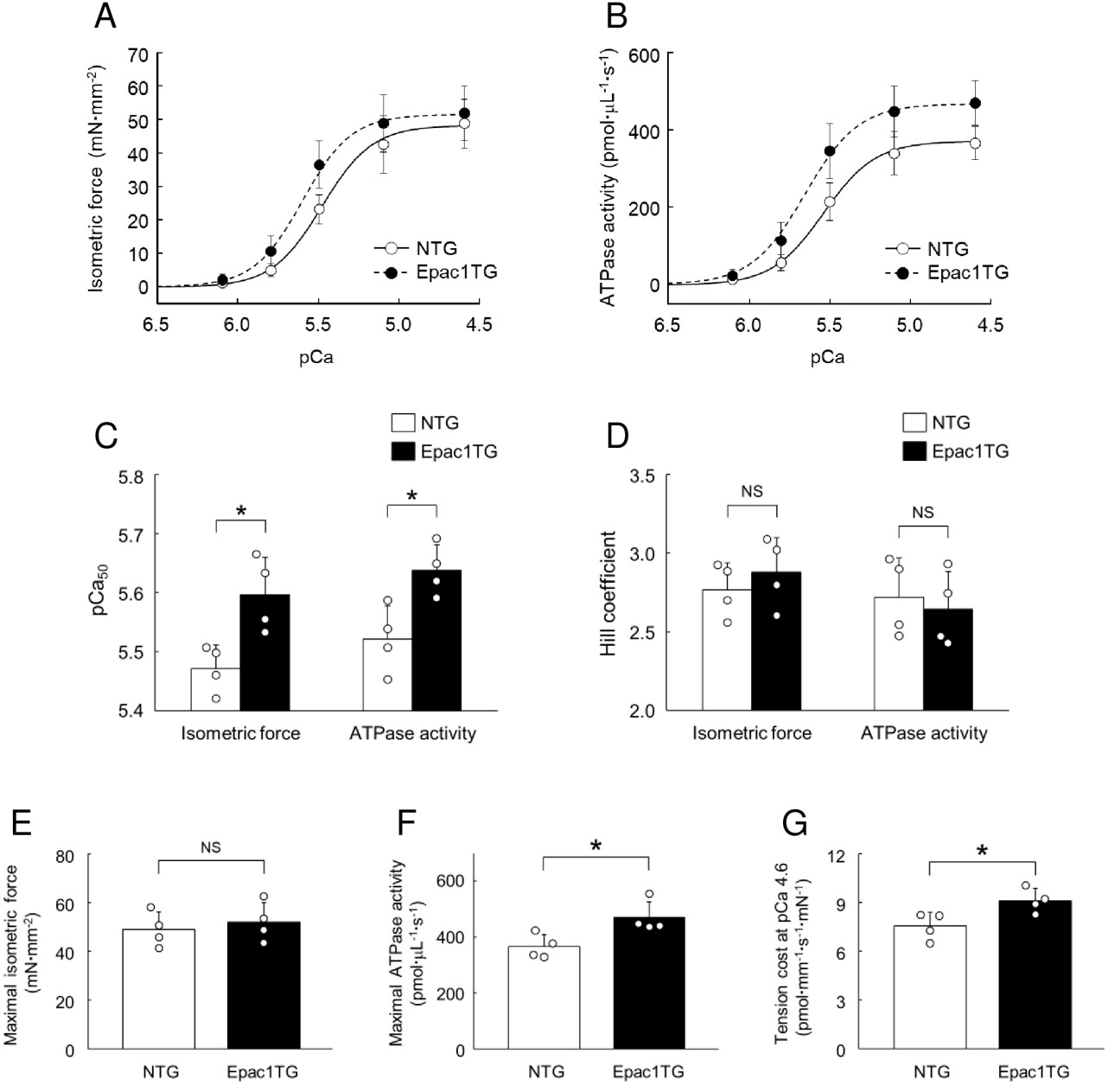

**Fig 6. Functional characterization of skinned myocardium prepared from Epac1TG.** (A, B) Average values of isometric force **(A)** and ATPase activity **(B)** at pCa 6.1, 5.8, 5.5, 5.1 and 4.6 were plotted for NTG (open circle) and Epac1TG (closed circle), and the data points were fitted to the Hill equation (solid line, NTG; dashed line, Epac1TG). (C) Average $pCa_{50}$ values ($Ca^{2+}$ concentration required for half maximal effect) of isometric force **(left)** and ATPase activity **(right)** were significantly greater in Epac1TG than in NTG (*$P<0.05$ by unpaired *t*-test). (D) Average Hill coefficient of isometric force **(left)** and ATPase activity **(right)** were similar in NTG and Epac1TG ($P=$NS by unpaired *t*-test). (E) Maximal isometric force at pCa 4.6 was similar in NTG and Epac1TG ($P=$NS by unpaired *t*-test). (F-G) Maximal ATPase activity **(F)** and tension cost at pCa 4.6 **(G)** were significantly greater in Epac1TG than in NTG (*$P<0.05$ by unpaired *t*-test). Bar blots represent means±SD and open circles show individual data from biological replicates of NTG and Epac1TG ($n=4$ each), each with 3 technical replicates.

## Maximal ATPase activity and tension cost are significantly greater in Epac1TG

We next compared the isometric force and ATPase activity at the maximal level of $Ca^{2+}$ activation (pCa = 4.6), i.e., maximal isometric force and maximal ATPase activity, in NTG and Epac1TG (**Fig 6E** **and** **6F**). Maximal isometric force was similar in NTG and Epac1TG (NTG vs. Epac1TG: 48.9 ± 7.2 vs. 52.0 ± 8.1 mN/mm², $P$ = NS by unpaired $t$-test) (**Fig 6E**). But, in contrast, maximal ATPase activity was significantly greater in Epac1TG than in NTG (NTG vs. Epac1TG: 367 ± 43 vs. 470 ± 57 pmol/µL/s, $P < 0.05$ by unpaired $t$-test) (**Fig 6F**). We also examined the average tension cost, i.e., ATPase activity/isometric force ratio, during force development at pCa 4.6 and this was also significantly greater in Epac1TG than in NTG (NTG vs. Epac1TG: 7.6 ± 0.8 vs. 9.1 ± 0.7 pmol/mm/s/mN, $P < 0.05$ by unpaired $t$-test) (**Fig 6G**).

These data suggest that the energetic cost of contraction might be increased in cardiac myofilaments of Epac1TG due to an imbalance between force-generating capacity and ATPase activity at maximal levels of $Ca^{2+}$ activation (pCa 4.6).

## Epac activation increases Ca²⁺ sensitivity of force and ATPase activity

The average values of isometric force and ATPase activity in skinned myocardium prepared from WT mice ($n$ = 6 each) were plotted with or without 8CPT-AM treatment (5 µM for 30 min) at the $Ca^{2+}$ concentration of 6.1, 5.8, 5.5, 5.1 and 4.6 (expressed as pCa = -log [$Ca^{2+}$]) (S8A and S8B Fig of S1 Data). The curves representing the force-pCa relationship (S8A Fig of S1 Data) and the ATPase activity-pCa relationship (S8B Fig of S1 Data) each showed a leftward shift (i.e., an increase of $Ca^{2+}$ sensitivity) in response to 8CPT-AM treatment, compared to the control, suggesting that activation of endogenous Epac increased $Ca^{2+}$ sensitivity of force and ATPase activity.

The $pCa_{50}$ of isometric force was significantly increased by 8CPT-AM treatment (Control vs. 8CPT-AM: 5.35 ± 0.07 vs. 5.48 ± 0.06, $P < 0.01$ by unpaired $t$-test) (S8C Fig of S1 Data), with no change of the Hill coefficient (Control vs. 8CPT-AM: 2.67 ± 0.40 vs. 2.81 ± 0.32, $P$ = NS by unpaired $t$-test) (S8D Fig of S1 Data). Similarly, the $pCa_{50}$ of ATPase activity was also significantly increased by 8CPT-AM treatment (Control vs. 8CPT-AM: 5.43 ± 0.09 vs. 5.55 ± 0.06, $P < 0.05$ by unpaired $t$-test) (S8C Fig of S1 Data), with no change of the Hill coefficient (Control vs. 8CPT-AM: 2.59 ± 0.30 vs. 2.73 ± 0.30, $P$ = NS by unpaired $t$-test) (S8D Fig of S1 Data).

These data all support the idea that activation of endogenous Epac increased $Ca^{2+}$ sensitivity of force and ATPase activity, compared to the control.

## Epac activation increases maximal ATPase activity and tension cost

We next compared the maximal isometric force and maximal ATPase activity in skinned myocardium with or without 8CPT-AM treatment (S8E and S8F Fig of S1 Data). Maximal isometric force was similar in the control and 8CPT-AM-treated skinned myocardium (Control vs. 8CPT-AM: 43.1 ± 7.0 vs. 44.6 ± 8.6 mN/mm², $P$ = NS by unpaired $t$-test). But, in contrast, maximal ATPase activity was significantly increased by 8CPT-AM treatment (Control vs. 8CPT-AM: 324 ± 54 vs. 392 ± 51 pmol/µL/s, $P < 0.05$ by unpaired $t$-test). We also examined the average tension cost, i.e., ATPase activity/isometric force ratio, during force development at pCa 4.6 and this was also significantly increased by 8CPT-AM treatment (Control vs. 8CPT-AM: 7.5 ± 0.8 vs. 8.9 ± 0.8 pmol/mm/s/mN, $P < 0.05$ by unpaired $t$-test) (S8G Fig of S1 Data).

Epac activation with 8CPT-AM in skinned myocardium increased the energetic cost of contraction, as in the case of Epac1TG (**Fig 6G**).

## Cardiac dysfunction is significantly greater in Epac1TG after chronic isoproterenol infusion

In order to examine the effects of the increased maximal tension cost and RLC phosphorylation on cardiac function under chronic catecholamine stress, we performed chronic ISO infusion (60 mg/kg/day for 1 week) in both Epac1TG and NTG and examined cardiac function by means of echocardiography (**Fig 7**).

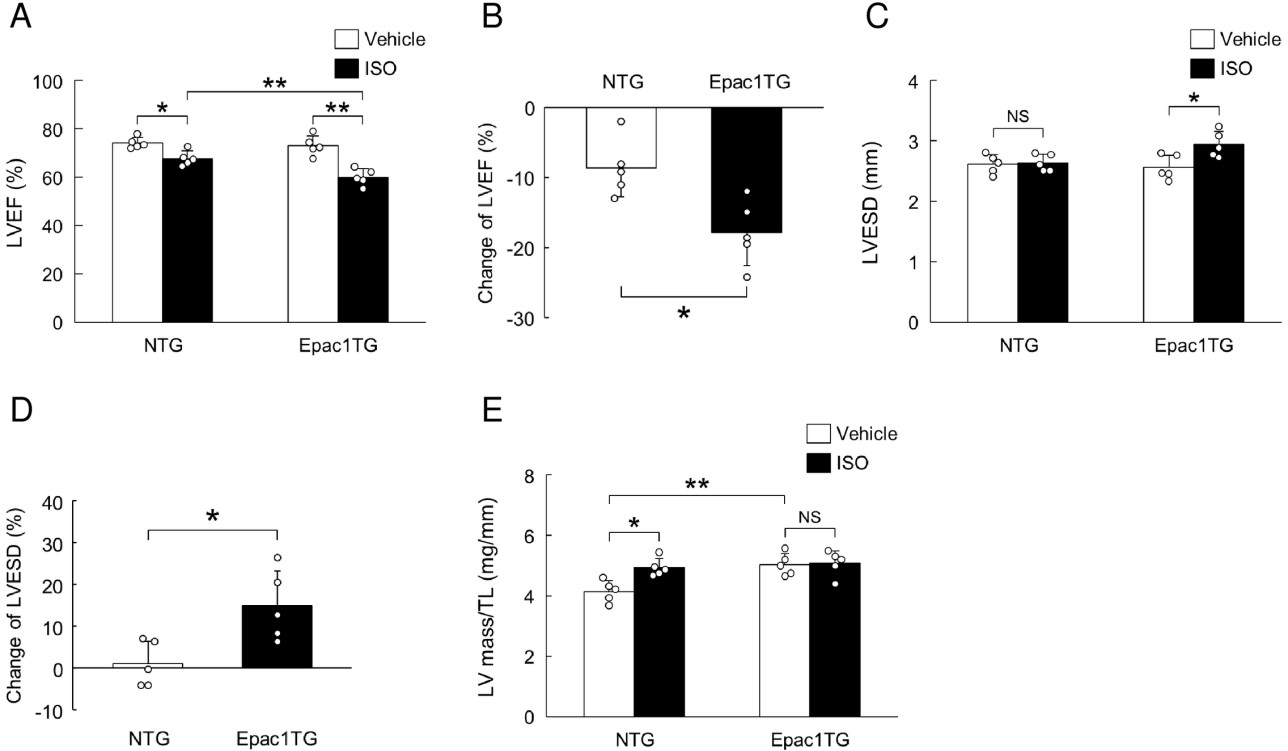

**Fig 7. Effects of chronic ISO infusion on cardiac function in Epac1TG.** (A) LVEF was similar between NTG and Epac1TG in the vehicle-treated group (*P* =NS by one-way ANOVA). LVEF in the ISO-treated group was significantly smaller than that in the vehicle-treated group in both NTG (*P<0.05 by one-way ANOVA) and Epac1TG (**P<0.01 by one-way ANOVA), but LVEF in ISO-treated Epac1TG was much smaller than that in ISO-treated NTG (**P<0.01 by one-way ANOVA). (B) Change of LVEF from the vehicle- to the ISO-treated group was significantly greater in Epac1TG than in NTG (*P<0.05 by unpaired *t*-test). (C) LVESD was similar between *t*he vehicle-treated group and the ISO-treated group in NTG (*P*=NS by one-way ANOVA), but was significantly greater in the ISO-treated group than in the vehicle-treated group in Epac1TG (*P<0.05 by one-way ANOVA). (D) Change of LVESD from vehicle- to ISO-treated group was significantly greater in Epac1TG than in NTG (*P<0.05 by unpaired *t*-test). (E) LV mass/tibial length ratio in vehicle-*t*reated Epac1TG was significantly greater than that in vehicle-treated NTG (**P<0.01 by one-way ANOVA). It was also significantly greater in the ISO-treated group than in the vehicle-treated group in NTG (*P<0.05 by one-way ANOVA), but remained unchanged and appeared to be saturated in Epac1TG (*P*=NS by one-way ANOVA). Bar blots represent means±SD and open circles show individual data from biological replicates of NTG and Epac1TG (*n*=5 each), each with 3 technical replicates.

The left ventricular ejection fraction (LVEF), taken as a measure of cardiac function, was not different between NTG with vehicle and Epac1TG with vehicle (NTG vs. Epac1TG; 74 ±2 vs. 73±4%, *P*=NS by one-way ANOVA, *n*=5 each) (**Fig 7A**). The LVEF in the ISO-treated group was significantly smaller than in the vehicle-treated group in both NTG (*P* <0.05 by one-way ANOVA) and Epac1TG (*P*<0.01 by one-way ANOVA) respectively and the magnitude of the decrease was much greater in Epac1TG than in NTG (NTG vs. Epac1TG; 68±3 vs. 60±3%, *P*<0.01 by one-way ANOVA, *n*=5 each) (**Fig 7A**). We also examined the decrease of cardiac function in response to chronic ISO in terms of % change of LVEF from vehicle-treated NTG (**Fig 7B**, **left)** and vehicle-treated Epac1TG (**Fig 7B**, **right)**, and it was also significantly greater in Epac1TG than in NTG (NTG vs. Epac1TG: −8.6±4.2% vs. −17.8±4.7%, *P*<0.05 by one-way ANOVA, *n*=5 each) (**Fig 7B**). Further, the left ventricular end-systolic diameter (LVESD) was not altered in the chronic ISO-treated group, compared to the vehicle-treated group in NTG (Vehicle vs. ISO; 2.61±0.16 vs. 2.63±0.14 mm, *P*=NS by one-way ANOVA, *n*=5 each), but it was significantly increased in Epac1TG (Vehicle vs. ISO; 2.56±0.20 vs. 2.94±0.21 mm, *P*<0.05 by one-way ANOVA, *n*=5 each) (**Fig 7C**). We also examined the increase of cardiac diameter in response to chronic ISO in terms of % change of LVESD from vehicle-treated NTG (**Fig 7D**, **left)** and vehicle-treated Epac1TG (**Fig**

[7D, right](), and it was significantly greater in Epac1TG than in NTG (NTG vs. Epac1TG: 0.9±5.4 vs. 14.8±8.4%, $P < 0.05$ by one-way ANOVA, $n = 5$ each) ([Fig 7D]()).

We next examined cardiac hypertrophy in response to chronic ISO infusion ([Fig 7E]()). The value of LV (mg)/tibial length (mm) ratio, taken as a measure of cardiac hypertrophy, was slightly but significantly greater in vehicle-treated Epac1TG than in vehicle-treated NTG ($P < 0.01$), as previously observed in vitro [44,55–57] and in vivo [21]. Chronic ISO infusion significantly increased the LV/tibial length ratio in NTG, but no change was seen in Epac1TG (NTG: from 4.1±0.4 [vehicle; $n = 5$] to 4.9±0.3 [ISO; $n = 5$], $P < 0.05$ by one-way ANOVA; Epac1TG: from 5.0±0.4 [vehicle; $n = 5$] to 5.1±0.4 mg/mm [ISO; $n = 5$], $P =$ NS by one-way ANOVA) ([Fig 7E]()).

These results suggest that the Epac1-mediated increase of tension cost and RLC phosphorylation in skinned myocardium might be important for the development of cardiac hypertrophy, and might play a pivotal role in cardiac remodeling in response to chronic catecholamine stress.

## Discussion

The effects of β-AR stimulation on the heart are mediated via activation of the cAMP/PKA pathway in myofilaments, principally targeting the thin filament protein TnI and the thick filament protein MyBP-C [22,58]. However, RLC phosphorylation is also increased by stimulation of $β_1$-AR, as well as TnI and MyBP-C phosphorylation [16]. In skinned myocardium, PKA-mediated phosphorylation of TnI and MyBP-C increases the rate of cross-bridge cycling and decreases the $Ca^{2+}$ sensitivity of force [4,59,60]. However, the mechanism of the RLC phosphorylation in response to β-AR stimulation and its physiological relevance in living myocardium remain poorly understood. We and the other groups have confirmed that the cAMP/PKA pathway does not alter RLC phosphorylation [22,58,61]. We therefore hypothesized that RLC phosphorylation might be increased via the cAMP/Epac1 pathway, and here, we examined this hypothesis using Epac1TG and WT mice treated with 8CPT-AM in vitro and in vivo.

Our results showed a leftward shift of the force-pCa relationship in Epac1TG compared to NTG, suggesting that Epac1 might increase the $Ca^{2+}$ sensitivity of force, as had been proposed based upon a study of permeabilized ventricular cardiac myocyte in vitro [25]. We also demonstrated that the $Ca^{2+}$ sensitivity of ATPase activity and the tension cost were significantly increased in Epac1TG, compared to NTG. Importantly, RLC phosphorylation was significantly and selectively increased in Epac1TG without affecting the phosphorylation status of other myofilament components, such as TnI and MyBP-C. Thus, it appears that the cAMP/Epac1 pathway induces selective phosphorylation of RLC, and this increases the $Ca^{2+}$ sensitivity of force and ATPase activity. Consequently, increased RLC phosphorylation via the cAMP/Epac1 pathway might lead to an imbalance between force-generating capacity and ATPase activity in living myocardium even in the basal state.

We also examined the effects of endogenous Epac1 activation with 8CPT-AM on the phosphorylation status of cardiac myofibril components in vitro using skinned myocardium and in beating heart prepared from WT mice and found that activation of endogenous Epac1 also might increase RLC phosphorylation without affecting the phosphorylation status of other myofilament components, and independently of PKA. More importantly, Epac-mediated RLC phosphorylation was mediated via PLC/PKC, not via previously identified pathways such as MLCK and ZIP kinase [8–11].

We compared the expression levels of Epac1 and its potential effectors such as PLCε, PKC, CaMKII, cMLCK and ZIPK in the intact myocardium and in skinned fibers of NTG and Epac1TG, and found that all these proteins were present in skinned fibers although their expression levels are lower than in intact myocardium (S1 Fig of [S1 Data]()), in accordance with previous findings by another group [25]. In other words, the skinning treatment did not remove all the membranes and other proteins, and thus Epac activation and its downstream signaling can take place in skinned myocardium.

We demonstrated that endogenous Epac activation increased phosphorylation of MYPT1 (Thr 696) and MYPT2 (Thr 646) via PLC/PKC, independently of PKA. Phosphorylation of MYPT results in loss of the ability of myosin phosphatase to dephosphorylate RLC [14] and its phosphorylation was reported to be regulated by either RhoA/Rho kinase [52] or

PKC-potentiated inhibitory protein of 17 kDa [62]. This work, together with the previous studies, demonstrated for the first time that RLC phosphorylation might be regulated by Epac via PLC/PKC.

These data also showed for the first time that Epac activation with 8CPT-AM increased RLC phosphorylation and inhibition of myosin light chain phosphatase activity via PLC/PKC signaling in skinned myocardium and in beating heart prepared from WT mice. Thus, it seems unlikely that Epac1-mediated RLC phosphorylation in Epac1TG is simply a non-specific consequence of other factors, such as cardiac hypertrophy.

The reason why chronic ISO-induced cardiac hypertrophy was not observed in Epac1TG might be related to the genetic background of Epac1TG and the β-AR desensitization mechanism. The genetic background of Epac1TG was FVB/NJ strain and the hypertrophic response to chronic ISO treatment in FVB/NJ was shown to be small, compared to other background mice [63,64]. We have previously confirmed that the magnitudes of the increase in response to chronic ISO infusion at the dose of 5 mg/kg/day for 7 days [63] and at the dose of 60 mg/kg/day for 7 days (this study) were similar and amounted to less than 10% increase over ISO treatment alone, which was much smaller than in mice with other genetic backgrounds [39]. Considering the 22% increase of LV hypertrophy in Epac1TG at baseline, the hypertrophic response to ISO might be saturated in Epac1TG at baseline. Also, chronic ISO infusion at the dose used in this study (60 mg/kg/day for 7 days) was demonstrated to decrease $\beta_1$-AR density by approximately 57% by us [38] and this might contribute to the decreased hypertrophic response to ISO, because the ISO-induced hypertrophic response is known to be mediated via $\beta_1$-AR [65].

Cazorla et al. demonstrated that Epac activation increases myofilament $Ca^{2+}$ sensitivity in permeabilized cardiomyocytes and the curves representing the tension-pCa relationship showed a leftward shift (i.e., an increase of $Ca^{2+}$ sensitivity) in permeabilized cardiomyocytes, as found in this study [25]. However, the findings regarding the phosphorylation status in cardiac myofibril proteins were different, i.e., phosphorylation of MyBP-C and TnI was increased via PLC/PKC/CaMKII in isolated adult rat cardiomyocytes treated with 8-CPT or infected with adenoviruses encoding human Epac1 protein. In our study, MyBP-C and TnI phosphorylation was not observed. The reason for the difference in myofilament phosphorylation status is unclear. However, the increased Epac-mediated RLC phosphorylation demonstrated in this study might be compatible with increased myofilament $Ca^{2+}$ sensitivity [5,66]. On the other hand, TnI phosphorylation is known to reduce myofilament $Ca^{2+}$ sensitivity, while the effect of MyBP phosphorylation on $Ca^{2+}$ sensitivity is controversial [increased [67], unchanged [68], or reduced [69]]. The phosphorylation of TnI or MyBP-C seems to be inconsistent with the Epac1-increased $Ca^{2+}$ sensitivity observed by us and by Cazorla et al [25].

We have previously demonstrated that CaMKII activation in the heart is similar in wild-type and Epac1-knockout mice [37], and thus CaMKII-mediated MyBP-C phosphorylation might not be induced by 8CPT-AM treatment, even though MyBP-C was previously demonstrated to be phosphorylated by CaMKII [70]. We also previously examined the effect of silencing each PKC isoform with siRNAs and found that Epac activation specifically activates PKCε without affecting the activities of other cardiac PKC subtypes such as PKCα, PKCβ, PKCγ and PKCδ [37]. More importantly, PKCα was demonstrated to phosphorylate MyBP-C, which leads to decreased myofilament $Ca^{2+}$ sensitivity and reduced contractility in myocytes [71,72]. These data are consistent with the idea that MyBP-C phosphorylation was not altered by treatment with PKC inhibitor BIM or CaMKII inhibitor KN-93, even though MyBP-C is known to be phosphorylated by PKC [71,72] or CaMKII [70].

In this study, Epac activation did not alter maximal isometric force. However, it was previously demonstrated that isolated skinned myocardium in exogeneous MLCK shows enhanced $Ca^{2+}$ sensitivity and maximal isometric force [5,66]. We showed that Epac1 activation induced RLC phosphorylation via PLC/PKC. Another group used phospho-peptide mapping to demonstrate that the PKC-mediated RLC phosphorylation site was distinct from the MLCK-mediated RLC phosphorylation site [12]. So far, PKC-mediated RLC phosphorylation has been reported to increase $Ca^{2+}$ sensitivity as in the case of MLCK, but the effect of PKC-mediated RLC phosphorylation on maximal isometric force remains to be established. Thus, the effects of Epac and MLCK on maximal isometric force might be different, i.e., Epac-mediated phosphorylation of RLC

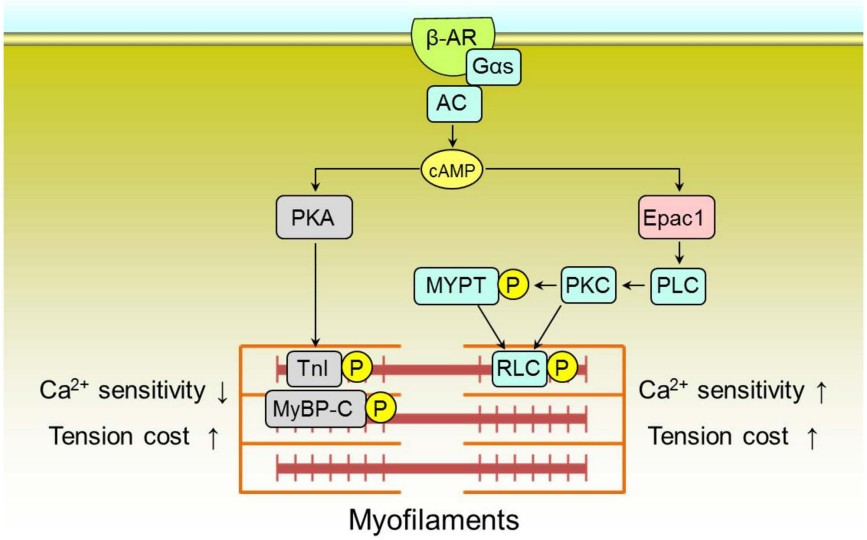

**Fig 8. Schematic summary of the proposed role of Epac1 in the regulation of myofilament function.** This scheme illustrates the proposed relationship between PKA (left) and Epac1 (right) in the regulation of Ca$^{2+}$ sensitivity and tension cost in cardiac myofilaments. Epac1-mediated increase of Ca$^{2+}$ sensitivity and decrease of tension cost represent findings in this study (blue-shaded columns) and PKA-mediated decrease of Ca$^{2+}$ sensitivity and increase of tension cost (dark-shaded columns) represent findings reported previously [24,73].

via PLC/PKC did not alter maximal isometric force, in accordance with the normal cardiac function of Epac1TG mice at baseline shown in **Fig 7**.

In order to examine the influence of the increased energetic cost of myofilament contraction on cardiac function, we next performed chronic ISO infusion in Epac1TG and NTG, and found that cardiac function was decreased more markedly in Epac1TG. This finding suggests that the Epac1-mediated RLC phosphorylation in cardiac myofilaments results in an increased oxygen consumption and an increased energetic cost of contraction at maximal levels of Ca$^{2+}$ activation.

In summary, we present the first evidence that activation of the cAMP/Epac1 pathway increases the Ca$^{2+}$ sensitivity of isometric force and ATPase activity, thereby increasing the tension cost (ATPase activity/force). This might increase the energetic cost of contraction and might be a first step in cardiac remodeling after catecholamine stress (**Fig 8**).

## Supporting information

**S1 Data. Supplementary figures.**
(PDF)

**S2 Data. Original images.**
(PDF)

## Author contributions

**Conceptualization:** Yoshiki Ohnuki, Yoshihiro Ishikawa, Satoshi Okumura.

**Formal analysis:** Yoshiki Ohnuki, Satoshi Okumura.

**Funding acquisition:** Yoshiki Ohnuki, Kenji Suita, Megumi Nariyama, Aiko Ito, Satoshi Okumura.

**Investigation:** Yoshiki Ohnuki, Kenji Suita, Misao Ishikawa, Yasumasa Mototani, Megumi Nariyama, Aiko Ito, Ichiro Matsuo, Yoshio Hayakawa, Akinaka Morii, Takao Mitsubayashi.

**Methodology:** Yoshiki Ohnuki, Yasutake Saeki, Satoshi Okumura.

**Supervision:** Yoshiki Ohnuki, Satoshi Okumura.

**Writing – original draft:** Yoshiki Ohnuki, Satoshi Okumura.

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
