## [Decision Letter · Decision Letter 0]

PONE-D-24-49143Epac1 increases myosin regulatory light-chain phosphorylation, energetic cost of contraction, and susceptibility to heart failurePLOS ONE

Dear Dr. Okumura,

Thank you for submitting your manuscript to PLOS ONE. After careful consideration, we feel that it has merit but does not fully meet PLOS ONE’s publication criteria as it currently stands. Therefore, we invite you to submit a revised version of the manuscript that addresses the points raised during the review process.

We look forward to receiving your revised manuscript.

Kind regards,

Pan Li, PhD

Academic Editor

PLOS ONE

Journal Requirements:

2. To comply with PLOS ONE submissions requirements, in your Methods section, please provide additional information regarding the experiments involving animals and ensure you have included details on methods of sacrifice.

3. Please provide the source of the animals used for the experiments

Reviewers' comments:

Reviewer's Responses to Questions

**Comments to the Author**

1. Is the manuscript technically sound, and do the data support the conclusions?

Reviewer #1: Yes

Reviewer #2: Partly

2. Has the statistical analysis been performed appropriately and rigorously? 

Reviewer #1: Yes

Reviewer #2: Yes

3. Have the authors made all data underlying the findings in their manuscript fully available?

Reviewer #1: Yes

Reviewer #2: Yes

4. Is the manuscript presented in an intelligible fashion and written in standard English?

Reviewer #1: Yes

Reviewer #2: No

5. Review Comments to the Author

Reviewer #1: This work by Ohnuki et al. dissects a novel function and potential pathogenic mechanism of Epac1 modulation of myosin regulatory light chain phosphorylation in the heart. The experimental approach is direct, and the data are reported clearly. I have several questions regarding the background levels of phosphorylation and how this may affect the data interpretation.

Comments.

Figure 4 shows myofilament phosphorylation levels with Epac activation and with PKA inhibition. The main result of this experiment shows that RLC phosphorylation is increased independent of PKA activity. However, it is strange that the other targets of PKA do not show any difference in phosphorylation. This suggests that the PKA targets are either already phosphorylated, and are unable to become dephosphorylated, or the PKA targets are at low level of phosphorylation. Can the authors comment of why PKA inhibition might not change known PKA phosphorylatin?

Figure 5 has a similar premise, activating Epac and then inhibiting specific kinases to establish a signaling cascade. MyBP-C is known to be phosphorylated by PKC and CaMKII, but its phosphorylation levels do not change.

Reviewer #2: The manuscript by Ohnuki et al. extends studies from their lab using a transgenic mouse that overexpresses the exchange protein activated by cyclic AMP, called Epac1. In this study they are trying to investigate the signalling pathways related to beta-adrenergic stimulation in the heart, and where Epac1 might play a role in altering myofilament protein phosphorylation, cardiac muscle force production, and cardiac muscle ATPase activity. The authors perform many biochemical, whole heart perfusion, echocardiography, skinned muscle fiber assays to investigate if activation of the cAMP/Epac1 pathway changes cardiac function. The authors suggest that Epac1 has a limited role in altering myofilament protein expression and protein phosphorylation levels, with the exception of regulatory light chain (RLC)—which it increase RLC phosphorylation. Epac1 also increases tension production, and ATPase, combining to increase the tension costs (i.e. the ATPase/Tension), and this brings some impairments or reductions in whole heart contractile function. These findings suggest that Epac1 might increase the energetic cost of contraction, leading to cardiac remodeling after catecholamine stress.

Detailed comments are listed below:

There are a lot of results in this study. The supplemental data is warranted and good to include, as it supports the primary figures. The organization of the results section seems a bit wandering, between biochemistry, then muscle function studies (roughly pages 14-16), then back to other biochemistry and additional muscle function studies (pages 19-20). While it appears there are initially the presentation of the transgenic Epac overexpression model, then in vitro activation of Epac, this is not clearly laid out as it is currently presented.

To this reviewer there is a concern or confusion about how Epac activation or signalling with 8CPT-AM takes place in skinned cardiac muscle fibers? There is some discussion about Epac1 being expressed in the cytosol and the nuclear ‘region’, but upon skinning all the membranes are removed as well as the other proteins, except for the myofilament proteins that remain. This is a large concern related to the interpretation of the study.

The listing of the n is not clear as to whether that is number of hearts or number of skinned fibers or number of biological vs. technical replicates in the case of the biochemical results.

There are some strange typographical or grammatical error in how results are listed or presented, such as:

-- page 13: “increase in Epac1TG was approximately 16-fold, compared to that in NTG (NTG (n = 6) vs. Epac1TG (n = 6): 100 ± 16 vs. 1,574 ± 244%, P < 0.01 by one-way ANOVA) (Fig 1B, left).” This has repeated listing of information and a strange use of parentheses.

-- page 16: “We prepared myocardium from hearts treated with (negative control) or without (positive control) Langendorff perfusion with calcium-free Tyrode’s solution for 1 hr, and examined the RLC”. This is not clear that is the negative and positive controls for what is perfused versus not perfused etc. Just unclear.

-- bottom of page 19: “significantly increased by 8CPT-AM treatment (Control (n = 10) vs. 8CPT-AM (n = 10): 5.35 ± 0.09 vs. 5.48 ± 0.08, P < 0.01 by unpaired t-test), with no change of the Hill coefficient (Control (n = 10) vs. 8CPT-AM (n = 10): 2.66 ± 0.48 vs. 2.79 ± 0.41, P = NS by unpaired t-test) (S1 Table of S1 Data).” This has an entirely different structure to try and foll compared to page 13, and is just a run on list of things to try and parse by the reader. This is confusing and unclear.

6. PLOS authors have the option to publish the peer review history of their article (what does this mean? ). If published, this will include your full peer review and any attached files.

**Do you want your identity to be public for this peer review?** For information about this choice, including consent withdrawal, please see our Privacy Policy .

Reviewer #1: No

Reviewer #2: No

---

## [Author Response · Author response to Decision Letter 1]

28 Feb 2025

Reviewer #1

This work by Ohnuki et al. dissects a novel function and potential pathogenic mechanism of Epac1 modulation of myosin regulatory light chain phosphorylation in the heart. The experimental approach is direct, and the data are reported clearly. I have several questions regarding the background levels of phosphorylation and how this may affect the data interpretation.

Comments.

Criticism-1:

Figure 4 shows myofilament phosphorylation levels with Epac activation and with PKA inhibition. The main result of this experiments shows that RLC phosphorylation is increased independent of PKA activity. However, it is strange that the other targets of PKA do not show any difference in phosphorylation. This suggests that the PKA targets are either already phosphorylated, and are unable to become dephosphorylated, or the PKA targets are at low level of phosphorylation. Can the authors comment of why PKA inhibition might not change known PKA phosphorylation?

Response-1:

In accordance with our results in this work, previous studies have found that phosphorylation levels of PKA targets such as MyBP-C and TnI at baseline are not altered by treatment with PKA inhibitor H-89 or protein kinase inhibitor peptide (PKI) [1, 2]. A possible explanation for the failure of PKA inhibitor to alter the basal phosphorylation level of MYBP-C or TnI is that the level of PKA-dependent phosphorylation might be too low for the effects of PKA inhibitors to be detectable [1, 2].

We incorporated the following sentence in the results section of the revised manuscript with new references (Page17, Lines 6-8).

Basal phosphorylation levels of PKA targets such as MyBP-C and TnI were not altered by H89, in accordance with the previous reports by other groups, probably due to the low levels of PKA-dependent phosphorylation [1, 2].

Criticism-2:

Figure 5 has a similar premise, activating Epac and then inhibiting specific kinases to establish a signaling cascade. MyBP-C is known to be phosphorylated by PKC and CaMKII, but its phosphorylation level do not change.

Response-2:

We have previously demonstrated that CaMKII activation in the heart is similar in wild-type and Epac1-knockout mice [3], and thus CaMKII-mediated MyBP-C phosphorylation might not be induced by 8CPT-AM treatment, even though MyBP-C was previously demonstrated to be phosphorylated by CaMKII [4]. We also previously examined the effect of silencing each PKC isoform with siRNAs and found that Epac activation specifically activates PKCε without affecting the activities of other cardiac PKC subtypes such as PKCα, PKCβ, PKCγ and PKCδ [3]. More importantly, PKCα was demonstrated to phosphorylate MyBP-C, which leads to decreased myofilament Ca2+ sensitivity and reduced contractility in myocytes [5, 6]. These data are consistent with the idea that MyBP-C phosphorylation was not altered by treatment with PKC inhibitor BIM or CaMKII inhibitor KN-93, even though MyBP-C is known to be phosphorylated by PKC [5, 6] or CaMKII [4].

We incorporated the above sentences in the discussion portion of the revised manuscript (Page 32, Lines 8-19).

Reviewer #2

The manuscript by Ohnuki et al. extends studies from their lab using a transgenic mouse that overexpresses the exchange protein activated by cyclic AMP, called Epac1. In this study they are trying to investigate the signaling pathways related to beta-adrenergic stimulation in the heart, and where Epac1 might play a role in altering myofilament protein phosphorylation, cardiac muscle force production, and cardiac muscle ATPase activity. The authors perform many biochemical, whole heart perfusion, echocardiography, skinned muscle fiber assays to investigate if activation of the cAMP/Epac1 pathway.changes cardiac function. The authors suggest that Epac1 has a limited role in altering myofilament protein expression and protein phosphorylation levels, with the activation of regulatory light chain (RLC)—which it increase RLC phosphorylation. Epac1 also increases tension production, and ATPase, combing to increase the tension costs (i.e. ATPase/Tesnsion), and this brings some impairments or reduction. These findings suggest that Epac1 might increase the energetic cost of contraction, leading to cardiac remodeling after catecholamine stress.

Detailed comments are listed below.

Criticism-1:

There are a lot of results in this study. The supplemental data is warranted and good to include, as it supports the primary figures. The organization of the results section seems a bit wandering, between biochemistry and additional muscle function studies (roughly pages 14-16), the back to other biochemistry and additional muscle function studies (page 19-20). While it appears there are initially the presentation of the transgenic Epac overexpression model, then in vitro activation of Epac, this is not clearly laid out as it is currently presented.

Response-1:

To improve the clarity, we have modified the layout by roughly dividing the results section into two parts (biochemistry data and muscle function studies) and then subdividing each into two sections (Epac1TG and in vitro activation with 8CPT-AM) as shown below. We think these changes make the results more accessible.

1) Biochemistry data

(1) Epac1TG: Fig. 1, Fig. S1, Fig. 2, Fig. S4 and Fig. 5A-C.

(2) 8CPT-AM: Fig. S2, Fig. 3, Fig. S3, Fig. S4, Fig. 4, Fig. S5, Fig. 5D-F and Fig. S6.

2) Muscle function studies

(1) Epac1TG: Fig. S7, Fig. 6, Fig. 7.

(2) 8CPT-AM: Fig. S8.

Criticism-2:

To this reviewer there is a concern or confusion about how Epac activation or signaling with 8CPT-AM takes place in skinned cardiac muscle fibers? There is some discussion about Epac1 being expressed in the cytosol and the nuclear ‘region’, but upon skinning all the membranes are removed as well as the other proteins, except for the myofilament proteins that remain. This is a large concern related to the interpretation of the study.

Response-2:

We compared the expression levels of Epac1 and its potential effectors such as PLCε, PKC, CaMKII, cMLCK and ZIPK in the intact myocardium and in skinned fibers of NTG and Epac1TG, and found that all these proteins are present in skinned fibers although their expression levels are lower than in intact myocardium (S1 Fig of S1 Data), in accordance with previous findings by another group [2]. In other words, the skinning treatment did not remove all the membranes and other proteins, and thus Epac activation and its downstream signaling can take place in skinned myocardium.

We incorporated the above sentences in the results portion of the revised manuscript (Page 30, Lines 8-14).

Criticism-3:

The listing of the n is not clear as to whether that is number of hearts or number of skinned fibers or number of biological vs. technical replicates in the case of the biochemical results.

Response-3:

We incorporated the required information in the revised manuscript.

Page 15, Lines 19-22

Page 16, Lines 6-7

Page 17, Lines 20-22

Page 18, Lines 11-15

Page 19, Lines 11-12

Page 21, Lines 5-6

Page 22, Lines 4-5

Page 23, Line 24-Page 24, Line 1

Criticism-4:

There are some strange typographical or grammatical error in how results are listed or presented , such as (Page 14, Lines 19-22):

---page 13: “increase in Epac1TG was approximately 16-fold, compared to that in NTG (NTG (n=6) vs. Epac1TG (n=6): 100 ± 16 vs. 1574 ± 244%, P < 0.01 by one-way ANOVA) (Fig 1B left). “This has repeated listing of information and a strange of parentheses.

Response-4:

We modified the sentences as shown below (Page 14, Lines 7-8).

---the magnitude of the increase in Epac1TG was approximately 16-fold, compared to that in NTG (P < 0.01 by one-way ANOVA) (Fig 1B, left).---

In addition, we modified parentheses in accordance with the Chicago Manual of Style {([ ])} in the revised manuscript.

Criticism-5:

---page 16: “We prepared myocardium from hearts treated with (negative control) or without (positive control) Langendorff perfusion with calcium-free Tyrode’s solution for 1 hr, and examined the RLC”. This is not clear that is negative and positive controls for what is perfused versus not perfused etc. Just unclear.

Response-5:

We modified the sentences as shown below (Page 15, Lines 19-22)..

We prepared skinned myocardium from 6 hearts perfused with a calcium-free Tyrode’s solution for 1 hr for the negative controls. For the positive controls, skinned myocardium was prepared from 6 hearts treated without perfusion.

Criticism-6:

---bottom of page 19: “significantly increased by 8CPT-AM treatment (Control (n = 10) vs. 8CPT-AM (n = 10): 5.35 ± 0.09 vs. 5.48 ± 0.08, P<0.01 by unpaired t-test) (S1 Table of S1 Data). “This has an entirely different structure to try and foll compared to page 13, and is just a run on list of things to try and parse by the reader. This is confusing and unclear.

Response-6:

We incorporated the S1 Table and S6 Fig of S1 Data in the original manuscript (in vitro activation data with 8CPT-AM) into S8 Figure of S1 Data in the revised manuscript, in the same manner as the Epac1TG data in Fig. 6. We hope this sufficiency clarifies the presentation.

References

1. Shaw EE, Wood P, Kulpa J, Yang FH, Summerlee AJ, Pyle WG. Relaxin alters cardiac myofilament function through a PKC-dependent pathway. Am J Physiol Heart Circ Physiol. 2009;297(1):H29-H36. https://doi: 10.1152/ajpheart.00482.2008 PMID: 19429819.

2. Cazorla O, Lucas A, Poirier F, Lacampagne A, Lezoualc'h F. The cAMP binding protein Epac regulates cardiac myofilament function. Proc Natl Acad Sci USA. 2009;106(33):14144-9. https://doi: 10.1073/pnas.0812536106 PMID: 19666481.

3. Okumura S, Fujita T, Cai W, Jin M, Namekata I, Mototani Y, et al. Epac1-dependent phospholamban phosphorylation mediates the cardiac response to stresses. J Clin Invest. 2014;124(6):2785-801. https://doi: 10.1172/jci64784 PMID: 24892712.

4. Flashman E, Redwood C, Moolman-Smook J, Watkins H. Cardiac myosin binding protein C: its role in physiology and disease. Circ Res. 2004;94(10):1279-89. https://doi: 10.1161/01.res.0000127175.21818.c2 PMID: 15166115.

5. Liu Q, Molkentin JD. Protein kinase Cα as a heart failure therapeutic target. J Mol Cellular Cardiol. 2011;51(4):474-8. https://doi: 10.1016/j.yjmcc.2010.10.004 PMID: 20937286.

6. Belin RJ, Sumandea MP, Allen EJ, Schoenfelt K, Wang H, Solaro RJ, et al. Augmented protein kinase C-α-induced myofilament protein phosphorylation contributes to myofilament dysfunction in experimental congestive heart failure. Circ Res. 2007;101(2):195-204. https://doi: 10.1161/circresaha.107.148288 PMID: 17556659.

---

## [Decision Letter · Decision Letter 1]

PONE-D-24-49143R1Epac1 increases myosin regulatory light-chain phosphorylation, energetic cost of contraction, and susceptibility to heart failurePLOS ONE

Dear Dr. Okumura,

Thank you for submitting your manuscript to PLOS ONE. After careful consideration, we feel that it has merit but does not fully meet PLOS ONE’s publication criteria as it currently stands. Therefore, we invite you to submit a revised version of the manuscript that addresses the points raised during the review process.

The reviewers acknowledge the authors' efforts in addressing previous comments but finds the reporting of biological and technical replicates unclear, raising concerns about the statistical and experimental rigor of the study. Clarification and potential additional experiments are necessary to ensure the validity of the findings.

We look forward to receiving your revised manuscript.

Kind regards,

Pan Li, PhD

Academic Editor

PLOS ONE

Reviewers' comments:

Reviewer's Responses to Questions

**Comments to the Author**

1. If the authors have adequately addressed your comments raised in a previous round of review and you feel that this manuscript is now acceptable for publication, you may indicate that here to bypass the “Comments to the Author” section, enter your conflict of interest statement in the “Confidential to Editor” section, and submit your "Accept" recommendation.

Reviewer #1: All comments have been addressed

Reviewer #2: (No Response)

2. Is the manuscript technically sound, and do the data support the conclusions?

Reviewer #1: Yes

Reviewer #2: No

3. Has the statistical analysis been performed appropriately and rigorously? 

Reviewer #1: Yes

Reviewer #2: Yes

4. Have the authors made all data underlying the findings in their manuscript fully available?

Reviewer #1: Yes

Reviewer #2: No

5. Is the manuscript presented in an intelligible fashion and written in standard English?

Reviewer #1: Yes

Reviewer #2: Yes

6. Review Comments to the Author

Reviewer #1: The authors have satisfactorily addressed my concerns with the additional points in the discussion.

Reviewer #2: The Authors have done a good job of addressing the reviewer responses. However, their number of biological replicates and number of technical replicates per biological replicate are still confusing. That is very concerning.

At a minimum, the authors need to have 4-5 biological replicates (hearts) for every experiment, and the experiments need to include all hearts among all the experiments. This is not clear. Then, within each biological replicate there needs to be at least 3 technical replicates per each heart--for every experiment or assay. It is clear there are some measurements that meet this guideline, but as written many of them do not. Thus, it is not enough information to statistically and experimentally represent the observations for the study design. As presented, it is a study design that is critically flawed and should not be published.

7. PLOS authors have the option to publish the peer review history of their article (what does this mean? ). If published, this will include your full peer review and any attached files.

**Do you want your identity to be public for this peer review?** For information about this choice, including consent withdrawal, please see our Privacy Policy .

Reviewer #1: No

Reviewer #2: No

---

## [Author Response · Author response to Decision Letter 2]

13 May 2025

Reviewer #1:

The authors have satisfactorily addressed my concerns with the additional points in the discussion.

Response:

Thank you.

Reviewer#2:

The Authors have done a good job of addressing the reviewer responses. However, their number of biological replicates and number of technical replicates per biological replicates are still confusing. That is very concerning.

At a minimum, the authors need to have 4-5 biological replicates (heart) for every experiment, and the experiments need to include all hearts among all the experiments This is not clear. Then, within each biological replicate there needs to be at least 3 technical replicates per each heart—for every experiment or assay. It is clear there are some measurements that meet this guideline, but as written many of them do not. Thus, it is not enough information to statistically and experimentally represent the observations for the study design. AS presented, it is a study design that is critically flawed and should not be published.

Response:

Thank you for your important comments. We apologize for the insufficient clarity regarding replicates, but should like to emphasize that the numbers of biological and technical replicates do in fact meet the necessary criteria. We have now clarified this in the text and figure legends as follows.

The numbers n of open circles in Fig 2, Fig 4, Fig 5, Fig 6, Fig S3, Fig S4, Fig S5 and Fig S8 in the original manuscript were larger than the numbers of biological replicates because we prepared two or more skinned fibers from one heart in some of the mice and the data from each skinned fiber from one heart were used for the analysis. We have revised the figures so that the number of open circles now corresponds to the number of biological replicates, each with 3 technical replicates that were averaged for further analysis.

We incorporated following sentences in the method sections of the revised manuscript (Page 13, Lines 1-3).

Open circles in the bar charts show individual data from biological replicates, each with 3 technical replicates that were averaged for subsequent analysis.

We also incorporated the following sentences in the figure legends of the revised manuscript.

in legends of Fig 1 (Page 48, Lines 15-16)

---Bar plots represent means ± SD and open circles show individual data from biological replicates of NTG and Epac1TG (n = 6 each), each with 3 technical replicates.

in legends of Fig 2 (Page 49, Lines 5-6)

---Bar plots represent means ± SD and open circles show individual data from biological replicates of NTG and Epac1TG (n = 4 each), each with 3 technical replicates.

in legends of Fig 3 (Page 50, Lines 1-3)

---Bar plots represent means ± SD and open circles show individual data from biological replicates of Control, 8CPT-AM, 8CPT-AM + H89 (n = 6 each), each with 3 technical replicates.

in legends of Fig 4 (Page 50, Lines 21-22)

---Bar plots represent means ± SD and open circles show individual data from biological replicates (n = 3 each), each with 3 technical replicates.

in legends of Fig 5 (Page 51, Lines 23-24)

---Bar plots represent means ± SD and open circles show individual data from biological replicates (n = 3 each), each with 3 technical replicates.

in legends of Fig 6 (Page 52, Lines 14-15)

---Bar plots represent means ± SD and open circles show individual data from biological replicates of NTG and Epac1TG (n = 4 each), each with 3 technical replicates.

in legends of Fig 7 (Page 53, Lines 12-13)

---Bar plots represent means ± SD and open circles show individual data from biological replicates of NTG and Epac1TG (n = 5 each), each with 3 technical replicates.

in legends of Fig S1 (S1 Data)

---Bar plots represent means ± SD and open circles show individual data from biological replicates of NTG and Epac1TG (n = 6 each), each with 3 technical replicates.

in legends of Fig S2 (S1 Data)

---Bar plots represent means ± SD and open circles show individual data from biological replicates (n = 6 each), each with 3 technical replicates.

in legends of Fig S3 (S1 Data)

---Bar plots represent means ± SD and open circles show individual data from biological replicates (n = 3 each), each with 3 technical replicates.

in legends of Fig S4 (S1 data)

---Bar plots represent means ± SD and open circles show individual data from biological replicates (n = 3 each), each with 3 technical replicates.

in legends of Fig S5 (S1 Data)

---Bar plots represent means ± SD and open circles show individual data from biological replicates (n = 3 each), each with 3 technical replicates.

in legends of Fig S6 (S1 Data)

---Bar plots represent means ± SD and open circles show individual data from biological replicates (n = 6 each), each with 3 technical replicates.

in legends of Fig S8 (S1 Data)

---Bar plots represent means ± SD and open circles show individual data from biological replicates (n = 6 each), each with 3 technical replicates.

---

## [Editor Report · Decision Letter 2]

Epac1 increases myosin regulatory light-chain phosphorylation, energetic cost of contraction, and susceptibility to heart failure

PONE-D-24-49143R2

Dear Dr. Okumura,

We’re pleased to inform you that your manuscript has been judged scientifically suitable for publication and will be formally accepted for publication once it meets all outstanding technical requirements.

Kind regards,

Pan Li, PhD

Academic Editor

PLOS ONE
---

## [Editor Report · Acceptance letter]

PONE-D-24-49143R2

PLOS ONE

Dear Dr. Okumura,

I'm pleased to inform you that your manuscript has been deemed suitable for publication in PLOS ONE. Congratulations! Your manuscript is now being handed over to our production team.

Kind regards,

on behalf of

Dr. Pan Li

Academic Editor

PLOS ONE